# A new method for estimating carbon dioxide emissions from drained peatland forest soils for the greenhouse gas inventory of Finland

Jukka Alm[1], Antti Wall[2], Jukka-Pekka Myllykangas[3], Paavo Ojanen[3], Juha Heikkinen[3], Helena M. Henttonen[3], Raija Laiho[3], Kari Minkkinen[4], Tarja Tuomainen[3], Juha Mikola[3]

[1]Natural Resources Institute Finland (Luke), Joensuu, Finland
[2]Natural Resources Institute Finland (Luke), Kokkola, Finland
[3]Natural Resources Institute Finland (Luke), Helsinki, Finland
[4]Department of Forest Sciences, University of Helsinki, Helsinki, Finland

*Correspondence to*: Jukka Alm (jukka.alm@luke.fi), Juha Mikola (juha.mikola@luke.fi)

**Abstract** In peatlands drained for forestry, the soil carbon (C), or carbon dioxide ($CO_2$) balance is affected by both (i) higher heterotrophic $CO_2$-C release from faster decomposing soil organic matter (SOM) and (ii) higher plant litter C input from more vigorously growing forests. This balance and other greenhouse gas (GHG) sinks and sources in managed lands are annually reported by national GHG inventories to the United Nations Climate Change Convention. In this paper, we present a revised, fully dynamic method for reporting the $CO_2$ balance of drained peatland forest soils in Finland. Our method can follow temporal changes in tree biomass growth, tree harvesting and climatic parameters, and is built on empirical regression models of SOM decomposition and litter input in drained peatland forests. All major components of aboveground and belowground litter input from ground vegetation and live and naturally died trees are included, supplemented by newly acquired turnover rates of woody plant fine roots. Annual litter input from harvesting residues is calculated using national statistics of loggings and energy use of trees. Leaching, which also exports dissolved C from drained peatlands, is not included. The results are reported as time series from 1990 to 2021 following the practice in GHG inventory. Our revised method produces an increasing trend of annual emissions from 0.2 to 2.1 t $CO_2$ ha$^{-1}$ yr$^{-1}$ for the period 1990–2021 in Finland (equal to a trend from 1.4 to 7.9 Mt $CO_2$ yr$^{-1}$ for the entire 4.3 Mha of drained peatland forests), with a statistically significant difference between years 1990 and 2021. Across the period 1990–2021, annual emissions are on average 1.5 t $CO_2$ ha$^{-1}$ yr$^{-1}$ (3.4 Mt $CO_2$ yr$^{-1}$ for 2.2 Mha area) in warmer southern Finland and -0.14 t $CO_2$ ha$^{-1}$ yr$^{-1}$ (-0.3 Mt $CO_2$ yr$^{-1}$ for 2.1 Mha area) in cooler northern Finland. When combined with data of the $CO_2$ sink created by the growing tree stock, the drained peatland forest ecosystems were in 2021 a source of 1.0 t $CO_2$ ha$^{-1}$ yr$^{-1}$ (2.3 Mt $CO_2$ yr$^{-1}$) in southern Finland and a sink of 1.2 t $CO_2$ ha$^{-1}$ yr$^{-1}$ (2.5 Mt $CO_2$ yr$^{-1}$) in northern Finland. We compare these results to those produced by the semi-dynamic method earlier used in the Finnish GHG inventory and discuss the strengths and vulnerabilities of the new revised method in comparison to more static emission factors.

## 1 Introduction

Drainage of peatlands improves forest growth but also turns unmanaged land to managed land and creates anthropogenic greenhouse gas (GHG) emissions: drainage increases aerobic decomposition of soil organic matter (SOM) and leads to substantial carbon dioxide ($CO_2$) emissions, globally accounting for 6% of total $CO_2$ emissions (Joosten 2010). Parties and signatories of the Paris Agreement and the United Nations Framework Convention on Climate Change (UNFCCC) are committed to reporting managed land GHG annual emissions and removals by sources and sinks, respectively from 1990 onwards. Emissions from drained peatlands are reported under the Land use, land-use change and forestry (LULUCF) sector of GHG inventories.

In Finland peatlands drained for forestry comprise an important category of managed lands. Almost one third, or 8.8 Mha (million hectares) of land area in Finland is covered by peatlands. This area also includes ca. 1 Mha thin-peated sites (< 30 cm peat layer). Of the total peatland area, 4.7 Mha is drained for forestry (Korhonen et al. 2021), 0.34 Mha for agriculture and 0.11 Mha for peat extraction (Statistics Finland 2022). Peatland forests have mainly been drained after World War II, with a peak in activity in 1960s and 1970s (Päivänen and Hånell, 2012). This paper addresses soil $CO_2$ emissions from those 4.3 Mha of drained peatland forests in Finland that fulfil the FAO definition of forest land. Soil $CO_2$ emissions from undrained peatland forests are excluded since the water table level, and thereby the functioning of soil, has not been modified by human activity. Emissions of $CO_2$ from drained peatland forest soils were first globally estimated by IPCC (2006). In Finland, the emissions, or soil C balance, have been quantified by Minkkinen et al. (2002) and Ojanen et al. (2014).

Countries may use static IPCC default emission factors for reporting soil $CO_2$ emissions from their drained peatland forest soils. Country-specific data and improvement of GHG inventory methods beyond the IPCC default emission factors are relevant especially for countries that have large areas of drained peatlands. Some countries have indeed opted to use country-specific emission factors; these can be based on the subsidence, i.e. fall of the surface of the organic soil, as in The Netherlands (National Institute for Public Health and the Environment 2021), $CO_2$ flux measurements as in Ireland (Duffy et a. 2021), or soil C balance modelling as in the UK (Brown et al. 2021). Weakness of static emission factors is that they cannot follow temporal changes, if such occur, in variables that drive the soil $CO_2$ balance. For instance, net $CO_2$ emissions from peatland forest soils may gradually evolve after drainage due to a successional transition of vegetation and changes in peat decomposition and hydrology (Sarkkola et al. 2010, Straková et al. 2012). In addition, increasing global temperatures likely increase SOM decomposition rates and thus soil $CO_2$ emissions.

Here we present a new, dynamic method for estimating the soil $CO_2$ balance of drained peatland forests in Finland, on time scales needed in the GHG inventory. We use both measured and modelled values of peat and litter decomposition and plant litter input. While water table depth (WTD) is the main factor that controls C loss from peat by heterotrophic respiration in

drained wetlands (Silvola et al. 1996, Ojanen et al. 2010, Jauhiainen et al. 2019), direct data of WTD is not nationwide available for GHG inventory purposes in Finland. We therefore link decomposition of peat and litter to tree basal area per hectare (BA) using empirical regression models available for different drained peatland forest site types (Ojanen et al. 2014). BA provides a proxy of the rate of evapotranspiration, which largely controls WTD in forestry-drained peatlands (Hökkä et al. 2021, Leppä et al. 2020). In addition, BA can be used to predict litter input from trees and ground vegetation (Ojanen et al. 2014). The time series of soil $CO_2$ balance is then calculated using a time series of BA provided by the Finnish National Forest Inventory, NFI (Tomppo et al. 2011, Korhonen et al. 2021), combined with a time series of air temperature. Since drained peatland forests are not evenly distributed across Finland, the temperature dependent peat and litter decomposition is modelled using spatially matched environmental variables. With this method, soil $CO_2$ balance can be estimated as a difference of C added to soil via litter input and $CO_2$ lost through SOM decomposition (Ojanen et al. 2012).

The $CO_2$ balance calculation method that was earlier applied in the Finnish GHG inventory for drained peatland forest soils also had dynamic components, but only in terms of belowground litter input. We call it as "semi-dynamic or previous method" in this paper. The estimate of SOM decomposition was static and spatially poorly representative as it relied on data collected from a limited number of field sites (Minkkinen et al. 2007). Aboveground litter input was not included in the previous calculations. Producing a revised, fully dynamic method became possible when new data and empirical models appeared that were able to link together the drivers and components of soil $CO_2$ balance (Ojanen et al. 2010, 2012, 2013, 2014). To illustrate the consequences of adopting the new method into the GHG inventory of Finland, we also present, for comparison, the $CO_2$ balance of drained peatland forest soils as estimated by the earlier semi-dynamic method (Statistics Finland 2022).

## 2 Materials and methods

### 2.1 The concept of soil $CO_2$ balance

The estimation of soil $CO_2$ balance ($CO_{2\,Net}$) is based on the annual difference between the release of $CO_2$ from decomposing soil organic matter (SOM), or heterotrophic soil respiration $R_{Het}$, and the C entering the soil through plant litter input, and can be expressed as:

$$CO_{2\,Net} = R_{Het} - \frac{44}{12}(I_{AGL} + I_{BGL} + I_{AGR} + I_{BGR}) \tag{1}$$

where $I_{AGL}$ and $I_{BGL}$ are the annual C mass input (t or Mt of C, as appropriate in the context) of aboveground and belowground litter, respectively, from living trees and ground vegetation; $I_{AGR}$ and $I_{BGR}$ are the annual C input of aboveground and belowground residues, respectively, from forest harvests and naturally dying trees, and 44/12 is the ratio of $CO_2$ and C molar masses, converting the organic C in dead organic matter to the respective mass unit of $CO_2$. Positive results in Eq. 1 appear when decomposition $R_{Het}$ is larger than the input of litter and residues and denote net $CO_2$ emission from soil

to the atmosphere. Negative values denote net increase of soil C stock. The time dimension of $CO_2$ balance is one year, and decomposition therefore includes wintertime decay as well.

$R_{Het}$ can further be defined as the sum of peat decomposition and decomposition of aboveground and belowground plant litter and residues from harvests and naturally died trees:

$$R_{Het} = D_P + D_{AGL} + D_{BGL} + D_{AGR} + D_{BGR} \qquad\qquad (2)$$

where $D_P$ is the annual $CO_2$ release (t or Mt of $CO_2$, as appropriate in the context) from decomposing peat, $D_{AGL}$ and $D_{BGL}$ are the annual $CO_2$ release from decomposing aboveground and belowground litter, respectively, of living trees and ground vegetation, and $D_{AGR}$ and $D_{BGR}$ are the annual $CO_2$ release from decomposing aboveground and belowground harvest and

natural mortality residues, respectively (Fig. 1). Autotrophic (root) respiration is not a part of the soil C stock change and is therefore not included in $R_{Het}$. To focus on $CO_2$ release from SOM, not confounded by autotrophic root respiration, the field plots were trenched (Ojanen et al. 2013).

## 2.2 Drained peatland forest site types

One key element in our method is that litter input and decomposition are individually estimated for each drained peatland

forest site type, hereafter denoted as "FTYPE". In Finland, FTYPE classification was developed for guiding drainage and forest management on peatlands. It follows the general Finnish forest site type theory (Cajander 1913) and uses species of trees and ground vegetation as indicators of moisture and fertility regimes. This typification of drained peatlands (Laine 1989, Laine et al. 2012) has been a practical tool for forestry and forest inventory in Finland. Forest growth and the rate of SOM decomposition are related to FTYPE: more fertile FTYPEs support greater tree biomass and growth, and consequently

also have higher decomposition rates (Silvola et al. 1996, Minkkinen et al. 2007, Ojanen et al. 2010).

The current typology of FTYPEs follows the principles presented in Laine et al. (2012) and includes a total of five FTYPEs: i.e., herb rich FTYPE, *Vaccinium myrtillus* FTYPE, *Vaccinium vitis-idaea* FTYPE, dwarf shrub FTYPE and *Cladonia* FTYPE, listed in order of decreasing soil fertility (Table 1). Differences in fertility between FTYPEs are evident in tree and ground vegetation species composition (Table 1). We did not separate the two subtypes of the *V. myrtillus* and *V. vitis-idaea*

FTYPEs (I and II with different original mire types; Laine et al. 2012), because field determination of especially *V. vitis-idaea* FTYPE subtypes can be ambiguous. Drained peatland forest sites have been classified into FTYPEs in field surveys in all NFIs, but only since NFI10 (carried out in 2004-2008) the classification, shown in Table 1, has been employed. The FTYPEs used in NFI8 (1986-1994) and NFI9 (1996-2003) were converted to correspond to the classification in Table 1 using field collected data on soil type, drainage status, site fertility and vegetation.

The areas and proportions of FTYPEs of all drained peatland "forests remaining forests" (i.e. forests that have not undergone another change in land use in past 20 years) in southern and northern Finland, derived from NFI12 (2014-2018), are shown

in Table 1 and the distribution of FTYPEs across Finland is illustrated in Fig. 2 (the definition of southern and northern Finland used in this paper follows the definition used in NFI). FTYPE areas have changed along the years due to changes in land use, such as new peatland drainage or reclamation of agricultural land. To provide a time series of FTYPE areas for the GHG inventory period starting from 1990 (Fig. 3), area data were collected from six NFIs, i.e., NFI8 (1986-1994), NFI9 (1996-2003), NFI10 (2004-2008), NFI11 (2009-2013), NFI12 (2014-2018) and NFI13 (2019- ). As for all other time series that utilize NFI data, the years between NFI mid-years are interpolated. However, the years 1990-1992 in northern Finland are extrapolated because the NFI8 mid-year was after 1990, and from 2017 onwards the values are determined with the help of the not yet complete data from NFI13. The development of Finnish NFI until NFI9 is described by Tomppo et al. (2011) and the sampling design of NFI12 by Korhonen et al. (2021).

The drained peatland forest sites included in our data are those classified as forests in the Finnish FAO Global Forest Resources Assessment (FRA) (FAO 2020): i.e. the area is ≥0.5 ha, trees are able to reach height of 5 m and the canopy cover is >10%. Currently, NFI field surveys include FRA land classification, and the earlier datasets from 1990s have been reclassified by NFI.

## 2.3 Climatic variables

The spatial distribution of FTYPEs (Fig. 2), and thereby their climatic attributes, differ due to those geographic reasons that originally resulted in the formation of different mires in Finland (Ruuhijärvi 1960, Havas 1961, Eurola 1962). Therefore, the mean May–October air temperature by FTYPE (approximating the growing season mean air temperature), which are needed to predict SOM decomposition in different FTYPEs (see Table 2), are calculated using real FTYPE distributions in southern and northern Finland (Fig. 2). Briefly, the May–October mean air temperature is first produced for each drained peatland forest location by combining the location data of sample plots in NFI12 with a 10 km × 10 km weather grid data provided by the Finnish Meteorological Institute (Venäläinen et al. 2005). Using the obtained data, FTYPE mean values are calculated for southern and northern Finland. For simplicity, NFI12 locations are used to calculate the FTYPE mean temperatures for all years in the 1990-2021 time series (i.e. we assume that FTYPE spatial distributions have not significantly changed in time). Lastly, the time series for both regions of the country are smoothed using 30-yr rolling means (Fig. 4). Smoothed long-term means follow long-term trends, such as gradual warming, but do not bring annual fluctuation in the results and thus help in revealing the anthropogenic impact of changes in drainage area, forest development and harvests on managed land emissions.

Decomposition, i.e. $CO_2$ production of litter from forest harvests and naturally died trees is calculated using the Yasso07 model (Tuomi et al. 2009, 2011), which uses mean annual temperature, temperature amplitude (the difference between the mean temperature of the warmest and coldest month divided by two), and annual precipitation as environmental drivers. Spatially weighed and smoothed 1990-2021 time series for these variables are produced as for May–October mean air temperature above, but since the harvest residue data is currently allocated between southern and northern Finland, not to

each FTYPE, the means are calculated for southern and northern Finland, not for each FTYPE (Fig. 4, Supplementary Fig.
1).

## 2.4 Basal area

FTYPE mean basal area per hectare, BA (Fig. 3) is used to predict peat and litter $CO_2$ release (see Table 2) and ground vegetation and arboreal fine root litter input (Tables 3, 4). The FTYPE mean BA is estimated using data of tree stem diameter (measured at breast height, DBH) of tree species groups (pine, spruce and deciduous species), collected in the
sample plots of the NFI8–NFI13 inventories. As in the case of FTYPE areas, FTYPE mean BAs are interpolated for years between the NFI mid-years, extrapolated for 1990–1992 in northern Finland and derived from the not yet complete NFI13 for years 2017–2021 to produce the time series needed in GHG inventory (Fig. 3). BAs have generally increased with time in drained peatlands, demonstrating the benefits of drainage for tree growth (Fig. 3).

## 2.5 Calculating litter input

Plant litter input consists of (1) litter from living trees (excluding fine root litter), (2) arboreal fine root litter (roots of trees and dwarf shrubs with ≤2 mm diameter), (3) ground vegetation litter (excluding dwarf shrub fine root litter) and (4) litter originating from forest harvests and natural tree mortality. In all calculations, all litter is assumed to contain 50% C of dry mass.

To estimate litter input from living trees, the mean biomass (Mg ha$^{-1}$) of each tree component (stem wood, stem bark, live
branches, dead branches, foliage, base/stump, and coarse roots of >10 mm diameter) for each FTYPE is first obtained from NFI. These biomasses are calculated by multiplying mean stem volume in each FTYPE with biomass conversion and expansion factors (BCEFs), based on biomass models of Repola (2008, 2009). BCEFs are component-biomass to stem-volume ratios, estimated using a sub-sample of trees measured in detail in each NFI for all different combinations of tree species group, FTYPE and South/North region. Inputs of litter from living trees, consisting of senesced foliage, branches,
stem and stump bark and coarse roots (>10 mm diameter), are then estimated by multiplying the biomass estimates and litter production rates (Table 5) of tree components. Litter production rate tells the proportion of the component mass that turns into litter in a year.

To calculate arboreal fine root litter input, fine root biomass is first estimated for each FTYPE by using FTYPE mean BAs of tree groups and FTYPE mean dwarf shrub cover (Table 6) as predictors in empirical regression models (Table 3), obtained
from Ojanen et al. (2014). Trees and dwarf shrubs are combined due to difficulties in distinguishing their fine roots in field samples. The dwarf shrub cover for the *Vaccinium myrtillus* and *Vaccinium vitis-idaea* FTYPEs are calculated as weighted means of values in I and II sub-types with the relative areas of sub-types (61.3% and 38.7% for the *V. myrtillus* FTYPE and 60.1% and 39.9% for the *V. vitis-idaea* FTYPE, respectively) as a weight factor. The fine root biomasses include an estimate for 0-20 cm rooting depth (Ojanen et al. 2013) plus 4.3% in deeper depths (Laiho and Finér 1996). Arboreal fine root litter

input is then estimated by multiplying fine root standing biomass by tree fine root turnover rate measured for each FTYPE (Table 6).

    Turnover rates of tree fine roots in Table 6 were estimated using minirhizotrons installed in each FTYPE (except for the *Cladonia* FTYPE, for which the estimate of the dwarf shrub FTYPE is used). Following the minirhizotron-method (Lukac 2012), transparent acrylic tubes (9–12 tubes per site) were installed into the soil on six sites to photograph fine roots

(Minkkinen et al., unpublished manuscript). After a one-year stabilisation period, all tubes were photographed 19 times during four consecutive years. It has been stated that minirhizotrons may underestimate root longevity in short (<3 year) studies since stabilization after installation disturbance may take several years (Strand et al. 2008). Our justification for a one-year stabilisation is that installation to peat soil causes little disturbance (no need to dig the soil) in comparison to installation to mineral soils, as in Strand et al. (2008). Images of 1.3 cm × 2 cm (height × width) were taken with a

minirhizotron camera (BTC-2; Bartz Technology, Santa Barbara, USA) and analyzed using a WinRHIZO Tron 2015b program (Régent, Canada). The diameters, lengths and depths of roots were recorded. Root longevities (1/turnover) were determined as years with the method by Kaplan and Maier (1958) using median longevity, i.e., the lifetime when 50% of the observed roots had died. If 50% mortality was not reached during the monitoring period, a parametric regression model with Weibull error distribution was applied to predict it using survreg (Kalbfleisch & Prentice 2002) in R-package Survival 3.4-0.

The turnover rates are based on tree fine roots with ≤0.5 mm diameter, and they are applied for all arboreal fine roots of ≤2 mm diameter.

    The mean combined aboveground and belowground litter input from ground vegetation (excluding dwarf shrub fine root litter input) is calculated for each FTYPE using FTYPE mean BA as a predictor in empirical regression models (Table 4), obtained from Ojanen et al. (2014). Notably, although dwarf shrub areal cover is assumed to remain static regardless of BA

in different FTYPEs (Table 3), the regression models that predict ground vegetation litter input (Tables 4, 5) have BAs of trees as predictors and thus follow changes in BA in FTYPEs.

    In forest harvests, most of the collected roundwood is utilized for wood products, but some is, together with harvest residues, used as energy wood. Therefore, to calculate litter input from harvest residues, a chain of calculations is needed. (1) Of all harvested roundwood volume in Finland (luke.fi/en/statistics/), the proportion that is collected from drained peatland forests

is first derived using the respective share of harvest area data provided by the NFI. (2) This proportion of roundwood volume is then converted to total tree biomass, and (3) using data provided by Luke Statistics (2022) of energy wood consumption, the part of biomass that is used for energy production is deducted from the total harvested biomass. (4) The remaining biomass is finally converted to different tree biomass components using specific BCEFs. Of these components, the foliage, branches, waste wood, stumps and coarse roots are assumed to remain in the harvest sites, and thus to compose the harvest

residue litter input.

    The yearly production of litter from naturally died trees is derived from NFI. For these trees, litter input consists of all biomass components, including the stem wood, calculated using appropriate BCEFs. Data of litter input from harvested and naturally died trees are currently available for southern and northern Finland, but not for each FTYPE separately.

Finally, litter input lacks one biomass component, i.e. tree roots in the diameter range of 2–10 mm. The reason is that there is neither biomass nor turnover rate estimate available for these roots. Omission of the 2-10 mm root litter input does not affect our estimate of heterotrophic decomposition of that litter fraction as it is based on inclusive empirical emission data, see below.

## 2.6 Calculating soil $CO_2$ release, $R_{Het}$

Annual $CO_2$ release from decomposing peat, $D_P$, and decomposing aboveground and belowground litter of living trees and ground vegetation, $D_{AGL}$ and $D_{BGL}$, is calculated using empirical regression models given in Ojanen et al. (2014). These models, presented in Table 2, include a FTYPE-specific constant and FTYPE May–October mean air temperature (Fig. 3) and FTYPE mean BA (Fig. 3) as dynamic predictors. The data behind the regression models include field measurements on 68 drained peatland forest sites (Ojanen et al. 2010, 2013), which cover the distribution of FTYPEs in Finland (see Fig. 1 in Ojanen et al. 2010). The measurements were originally performed on field plots with aboveground litter removed, but the $CO_2$ release from decomposing aboveground litter was later calculated using the Yasso07 decomposition model and added to the recorded values of $CO_2$ release (Ojanen et al. 2013). To focus on $CO_2$ release from SOM, not confounded by autotrophic root respiration, the field plots were trenched (Ojanen et al. 2013). The predicted $CO_2$ release is a yearly estimate although the temperature predictor is the mean May–October air temperature (Ojanen et al. 2014).

The soil $CO_2$ release estimates produced by the regression models in Table 2 do not include $CO_2$ release from harvest residues (the empirical data for the models was collected at sites without recent harvests) or from stump and stem wood of naturally died trees (because these parts of litter did not fit inside the $CO_2$ flow measurement chambers). The $CO_2$ release from these litter components is therefore estimated using the Yasso07 decomposition model, combined with time series of southern and northern Finland mean annual air temperature, temperature amplitude and mean annual precipitation (Fig. 4, Supplementary Fig. 2). Since the Yasso07 model produces the remaining OM pool after decomposition, the $CO_2$ release from decomposing aboveground and belowground harvest residues and stump and stem wood of recently naturally died trees, $D_{AGR}$ and $D_{BGR}$, is calculated by subtracting the remaining OM pool from the inputs of these residues, $I_{AGR}$ and $I_{BGR}$.

The Yasso07 model was first initialized for 50 years using constant residue input from harvests and naturally died trees (mean of 1970-1976 input for southern Finland and mean of 1975-1977 input for northern Finland). The initialized model was then run from 1971 onwards using real annual residue input. During the 50-yr initialization, the OM pool originating from harvest residues and naturally died trees was monotonically approaching a plateau (Supplementary Fig. 2). The rationale for using unusually short spin-up is that there is no long history for drained peatland forests in Finland (drainage activity rose after 1940s and was most intensive in 1970s) and the soil OM pool originating from anthropogenic activity has not been developing for long. Running longer spin-up would create unnaturally high harvest and natural tree mortality-based soil OM pool, leading to erroneously high $CO_2$ release from these, slowly decomposing C stocks during the GHG inventory reporting period. However, to be aware of the effects of the short spin-up, we also initialized the Yasso07 model for 1000

years. Increasing the length of initialization from 50 to 1000 years increased $CO_2$ release along the years 1990–2021 annually by 2.4–0.6% (mean 1.4%) and 4.1–1.5% (mean 2.8%) in southern and northern Finland, respectively.

### 2.7 Uncertainty analysis

Uncertainty was assessed for each annual estimate of soil $CO_2$ balance, as well as for the estimate of change in balance between years 1990 and 2021. We use the IPCC (2006) Guidelines convention, where uncertainty is defined as $1.96 \times$ S.E.M. and given as a percentage of the sink/source estimate. When the uncertainty of the estimate is less than 100 %, zero is not included within the 95 % confidence limits of the estimate. The accounted sources of uncertainty included NFI sampling errors in area and BA estimates, estimation error in the parameters of the models, litter production rates and dwarf shrub coverage, and uncertainty about litter production originating from living trees, harvests, and natural mortality. The details of variance propagation are given in the Appendix 1. These and all other computations were carried out in the R environment (R Core Team 2020).

### 2.8 Earlier version of the calculation method

The earlier version of the calculation method of drained organic forest soil $CO_2$ emissions applied in the GHG inventory of Finland (Statistics Finland 2022) differs in many ways from the new revised method we present here. In the earlier version, the estimates of heterotrophic soil respiration in FTYPEs are based on results from two peatlands, located at 61°N 25°E (covering FTYPEs from the herb rich to dwarf shrub FTYPE) and 62°N 31°E (representing the *Cladonia* FTYPE) (Nykänen et al. 1998, Minkkinen et al. 2007). The estimates include $D_P$ and $D_{BGL}$ only, are static in time and do not make a difference between southern and northern Finland. Belowground litter input from living trees, ground vegetation, harvest residues and natural mortality is included, whereas aboveground litter input (and most of its decomposition) from living plants and harvests is excluded. Also, in contrast to the new revised method, the belowground litter input from ground vegetation is constant, 1.1 t $CO_2$-C (or 4.0 t $CO_2$) ha$^{-1}$ yr$^{-1}$ (Laiho et al. 2003), the arboreal fine root turnover rate is 0.85 for all FTYPEs (Liski et al. 2006), and tree leaf biomass is used as a proxy to estimate arboreal fine root biomass (Helmisaari et al. 2007).

### 3 Results

### 3.1 Soil CO₂ balance

The method developed in this study produces a generally increasing trend of annual emissions from 0.2 to 2.1 t $CO_2$ ha$^{-1}$ yr$^{-1}$ for the period 1990–2021, equal to a trend from 1.4 to 7.9 Mt $CO_2$ yr$^{-1}$ for the entire 4.3 Mha area of drained peatland forests in Finland (Fig. 5). The difference between years 1990 and 2021 is statistically significant (uncertainty 46%; Table A10 in Appendix A). Across the period 1990–2021, emissions are on average 1.5 t $CO_2$ ha$^{-1}$ yr$^{-1}$, or 3.4 Mt $CO_2$ yr$^{-1}$ for the whole area of 2.2 Mha in southern Finland and -0.14 t $CO_2$ ha$^{-1}$ yr$^{-1}$, or -0.3 Mt $CO_2$ yr$^{-1}$ for the whole area of 2.1 Mha in northern

Finland (Fig. 5). Considering the uncertainty, drained organic forest soils in southern Finland and in the whole country are a source of $CO_2$ in 2021, while the 2021 net emission in northern Finland does not significantly differ from zero (Table A9 in Appendix A).

In contrast to the results produced by the new method, the earlier method produces a decreasing trend of total emissions from 12.1 to 3.2 Mt $CO_2$ for the period 1990–2021 for the whole country and the emissions are on average lower in southern than northern Finland (Fig. 5).

### 3.2 Total litter input and decomposition

The new method produces an increasing trend of litter input and decomposition for both southern and northern Finland (Fig. 6). In 2021, total litter input and decomposition are, per unit area, 28% and 34% higher in southern than northern Finland, respectively (Fig. 6). In both regions, the increasing trend in total litter input is mainly due to increasing input of aboveground tree litter and belowground arboreal litter (Fig. 6). These litter fractions also include residues from forest harvests and naturally died trees.

When compared to the earlier method, the new method produces 37% and 61% smaller belowground arboreal litter input, per unit area, for southern and northern Finland, respectively (Fig. 6). However, total litter input is on average 62% and 32% higher, respectively, because aboveground tree litter and ground vegetation litter are included in the new method (Fig. 6). In 2021, the new method produces 63% and 10% higher $CO_2$ release from decomposition, per unit area, than the previous method for southern and northern Finland, respectively (Fig. 6).

### 3.3 Input and decomposition of litter from harvested and naturally died trees

With the new method the input of litter from harvested and naturally died trees, as well as the $CO_2$ release from their decomposition, are significantly higher, per unit area, in southern than northern Finland but increase with time in both regions (Fig. 7). The $CO_2$ balance of these residues is negative (thus acting as a sink) and has a decreasing overall trend in both regions (Fig. 7). In 2021, the residues form 20% and 15% of all litter input, and their decomposition 15% and 12% of total decomposition in southern and northern Finland, respectively (compare Figs. 6 and 7).

### 3.4 Comparison of FTYPEs

When litter input and decomposition (excluding litter from harvested and naturally died trees) are estimated with the new method for each FTYPE individually, both litter input and decomposition have a general decreasing trend along the FTYPE fertility gradient from the most fertile herb rich FTYPE (Rhtkg) to the least fertile *Cladonia* FTYPE (Jatkg) (Fig. 8). The only exception in this pattern is litter input in the *Vaccinium vitis-idaea* FTYPE (Ptkg), which in 2021 exceeds litter input in

the more fertile *Vaccinium myrtillus* FTYPE (Mtkg) by 26% and 20% in southern and northern Finland, respectively (Fig. 8).

No clear trend of $CO_2$ balance emerges along the FTYPE fertility gradient (Fig. 8). Instead, net emissions are higher in both ends of the gradient than in the *Vaccinium vitis-idaea* (Ptkg) and the dwarf shrub (Vatkg) FTYPEs (Fig. 8). In these two FTYPEs, ground vegetation litter input and/or arboreal fine root litter input are relatively high in comparison to other FTYPEs (Fig. 8). In general, the share of ground vegetation litter of total litter input is higher in northern than southern Finland and increases towards the nutrient poor end of the FTYPE fertility gradient (Fig. 8).

### 3.5 Sensitivity of soil $CO_2$ balance to the main dynamic drivers

As temperature, T (and other climatic variables in Yasso07 modelling) and BA are the main dynamic drivers of soil $CO_2$ balance in our method, we examined their relative importance by producing three sets of scenarios, where either (1) climate, (2) BA (and the associated harvest rate), (3) both or (4) neither (the default method) were fixed to their 1990 values (Fig. 9): the first set is for the $CO_2$ balance of tree and ground vegetation litter input and SOM decomposition (top row of graphs, excluding harvest residues and litter from naturally died trees), the second set is for the $CO_2$ balance of input and decomposition of litter from harvested and naturally died trees (middle row) and the third set is for the total soil $CO_2$ balance (bottom row).

The $CO_2$ balance of tree and ground vegetation litter input and SOM decomposition (Fig. 9 top row) appears to be sensitive to the increasing T as the scenario with constant BA (i.e. only T increases) closely follows the default scenario. The effects of changes in BA are more subtle: increasing BAs until ca. 2010 (Fig. 3) decrease emissions slightly as the scenario with constant climate (only BA changes) produces less $CO_2$ than the scenario where both BA and climate remain constant. From ca. 2010 onwards BAs in FTYPES level off or start decreasing (Fig. 3) and the BA effect turns neutral (northern Finland) or slightly positive (southern Finland) (Fig. 9 top row).

The $CO_2$ balance of harvest residues and litter from naturally died trees (Fig. 9 middle row) is driven by changes in BA and harvest rate as the scenario with constant climate (only BA and harvest rate change) closely follows the default scenario. Changes in climate increase emissions slightly as the scenario with constant BA and harvest rate (only climate changes) produces more $CO_2$ than the scenario where BA, harvest rate and climate all remain constant.

The scenarios for the total soil $CO_2$ balance (Fig. 9 bottom row) combine the main characteristics of the scenarios for the two parts of the total balance: i.e. that the shape and the positive slope of $CO_2$ emissions in the default scenario are driven by climate warming, and that the main result of increasing BA and harvest rate from 1990 onwards is a lower level of net soil $CO_2$ emissions.

## 3.6 Ecosystem $CO_2$ balance

The ecosystem $CO_2$ balance in drained peatland forests is governed by the combination of soil $CO_2$ balance and the $CO_2$ sink created by living trees. When the soil $CO_2$ balance results of the new method are combined with tree $CO_2$ sink for southern Finland, the ecosystem $CO_2$ balance is negative, i.e. the forests are a $CO_2$ sink, until the beginning of 2010s when the balance turns positive (Fig. 10). This change of pattern is a product of both lower tree $CO_2$ sink in 2010s and increasing soil $CO_2$ emissions (Fig. 10). In northern Finland, drained peatland forests are still a $CO_2$ sink, but also there the gradually decreasing tree $CO_2$ sink together with increasing soil $CO_2$ emissions creates a decreasing trend for the ecosystem sink in 2010s (Fig. 10).

## 4 Discussion

### 4.1 The $CO_2$ balance of drained organic soils by the new method

The $CO_2$ balance of drained organic soils in Finland, as estimated by the new method we describe in this paper, has two main features (Fig. 5): first, drained peatland soils are currently a source of $CO_2$ in Finland, and second, the emissions are increasing with time due to the high sensitivity of SOM decomposition to increasing temperature. The mean increase of ca. 0.7 °C in mean May-October temperatures across the time series in FTYPEs in both southern and northern Finland (Fig. 4) increase CO2 emissions from soil by 8.1 Mt CO2 in the whole country (the constant BA and harvest scenario in Fig. 9 bottom row). An increase in BA and harvest rate over the decades counteract this pattern (the constant climate scenario in Fig. 9 bottom row), however, and the increase in net soil $CO_2$ emissions across the time series remains smaller (the default scenario in Fig. 9 bottom row).

Both litter input and decomposition have increasing trends in time (Fig. 6). The increasing trend in litter input is mainly driven by the general increase of BA across the FTYPEs (Fig. 4), which leads to greater litter input from living trees (evident in both aboveground tree litter input and belowground arboreal litter input in Fig. 6). Part of the increasing trend is also explained by the increasing input of litter from harvested and naturally died trees (Fig. 7). Litter input from ground vegetation, instead, remains rather stable across the time series (Fig. 6). The increasing $CO_2$ release from decomposition is driven by both increasing BA and increasing May-October temperature (Fig. 3) as both are positively associated with decomposition in the empirical data (Table 2; Ojanen et al. 2014). Part of the increasing trend is also explained by the increasing $CO_2$ release from the decomposing litter of harvested and naturally died trees (Fig. 7). The likely reasons why BA is positively linked to decomposition in the regression models by Ojanen et al. 2014 (Table 2) is that increasing litter input sustains larger decomposer biomass and thus higher $CO_2$ production, and that higher BA - through higher evapotranspiration – maintains lower soil water table level and a deeper oxic layer for SOM decomposition (Sarkkola et al. 2010). When the impacts on litter input and decomposition are combined, increasing BA first slightly decreases net soil $CO_2$ emissions

(scenario of constant climate in Fig. 8 top row), but this effect disappears in mid 2010s, when the positive trend in BA levels off or turns negative in most FTYPEs (Fig. 3).

Release of $CO_2$ from decomposition per area of drained peatland forest soil is significantly higher in southern than northern Finland (Fig. 6). This is due to higher temperature, but also due the generally higher BA and litter production (Fig. 3, 4, 6) in southern Finland. Input of litter from forest harvests has an increasing trend, and in the timescale of the inventory time series, harvest residues create a $CO_2$ sink and decrease net soil $CO_2$ emissions (Fig. 7). However, this does not imply that net ecosystem $CO_2$ emissions would decrease with harvests because tree $CO_2$ sink simultaneously decreases (see trends in 2010s in Fig. 10).

Of the FTYPEs, the dwarf shrub (Vatkg) and *V. vitis-idaea* (Ptkg) FTYPEs are clustered to the water divides (Suomenselkä and Maanselkä) of the southern and northern regions of the country, while other FTYPEs are more evenly distributed across the country (Fig. 2). These two common FTYPEs have relatively low nutrient status (Table 1) and both have, even without the harvest residue impact, for most part of the GHG inventory time series a negative or near zero soil $CO_2$ balance in the North, and the *V. vitis-idaea* FTYPE also in the South (Fig. 8). This is roughly in line with earlier findings (Minkkinen et al. 1999, Ojanen et al. 2014). However, the ubiquitous upwards temporal trend of soil $CO_2$ balance in all FTYPEs (Fig. 8) suggests that also these FTYPEs are entering the phase of net soil $CO_2$ emissions.

Uncertainty due to model parameters and input data were propagated into the soil $CO_2$ balance estimates. The parameter estimates of the peat and litter decomposition model were clearly the greatest source of uncertainty, followed by those in the fine-root litter input model (including the turnover rates) and litter input from living trees. Meanwhile, uncertainty in NFI estimates of site type areas and BA had negligible impact on annual balance estimates and somewhat greater, but still minor, impact on the estimates of changes in emissions. This indicates that the proposed method might also be applicable in somewhat smaller regions (with less precise NFI estimates) without drastic increase in uncertainty. Errors in annual soil $CO_2$ balance estimates are strongly correlated over the whole time series because the same model parameters with the same estimation errors are applied throughout the series. Since all applied models were linear in both their parameters and inputs, it was possible to handle these correlations by combining quantities including the same parameter (Appendix A). Uncertainty propagation by means of variances relies on an assumption of symmetric error distributions. This assumption is realistic since uncertainty due to all individual components in the balance estimates was 30% or less, i.e., coefficient of variation was at most 0.15 (Table A9; cf. Frey et al. 2006, sec. 3.2.2.4). Structural uncertainty about the models could not be accounted for since large-scale validation data does not exist.

## 4.2 Ecosystem $CO_2$ balance in drained peatland forests

Peatlands in Finland were mostly drained for enhancing tree growth and thus enabling forestry on peatlands. The increasing BAs in drained peatlands from 1990 onwards (Fig. 4) show that this target has largely been achieved. Therefore, when considering the net climate effects of peatland drainage, the development of tree $CO_2$ sink needs to be taken into consideration (Fig. 9). In 2010s, the trend of gradually increasing BA in drained peatland forests levelled off (Fig. 3). This is

due to increasing harvests in these forests (manifested by the increasing harvest residue input in Fig. 7), which in turn is associated with the sharp turn in tree $CO_2$ sink and net ecosystem balance after 2010, ultimately leading to a shift from a net

ecosystem sink to a source of $CO_2$ in southern drained peatland forests (Fig. 9). After 2010, drained peatland forests have been a sink of $CO_2$ only in northern Finland, but when the northern tree stands mature for harvesting, and industry demand for wood remains high, the net ecosystem $CO_2$ sink will likely be lost in northern Finland as well. Calculated for the whole country, forests growing on drained peatlands were a net sink of 0.2 Mt $CO_2$ in 2021.

## 4.3 Comparing the soil $CO_2$ balance predictions by the new and earlier method

The differences in the estimates of soil $CO_2$ balance between the new fully dynamic and the earlier semi-dynamic method are remarkable (Fig. 4). The main difference are the opposite trends in time series: in the beginning of the GHG inventory reporting period the new method produces 10.7 Mt lower soil $CO_2$ balance than the earlier method, but then reaches 3.4 Mt higher balance in recent years. The mean level of emissions differs a lot in the North, less in the South (Fig. 5).

The contrasting trends of time series produced by the two methods call for explanations. In the earlier method, $CO_2$ release

from decomposition is constant, whereas in the new method $CO_2$ release from decomposition increases in time as it is driven by increasing temperatures and BA (Fig. 6). Both methods predict increasing litter input in time due to increasing BA (Fig. 6). In the earlier method, however, increasing litter input combined with constant decomposition unavoidably leads to decreasing net emissions as the release of $CO_2$ in decomposition does not follow the increase in litter input. In the new method, both constituents of the $CO_2$ balance are dynamic, and the method can produce either an increasing or decreasing

trend of net emissions depending on the relative magnitude of slopes in trends for litter input and decomposition. The new method predicts that decomposition increases more in time than litter input does, thus leading to an increasing trend in net soil $CO_2$ emissions.

Besides the difference in the estimation of decomposition, estimation of litter input has major differences between the methods. First, while the above-ground litter (including litter from living trees and ground vegetation, as well as from

harvested and naturally died trees) is explicitly included in both litter input and decomposition calculations in the new method, the earlier method assumes that above-ground litter input and decomposition are in equilibrium, and consequently, above-ground litter is not included in the calculation. However, findings from drained organic forest soils suggest that while increasing litter input is generally associated with increasing decomposition, litter input and decomposition are seldom in equilibrium. For instance, in Finland, litter production exceeds decomposition in the nutrient-poor FTYPEs and vice versa in

the nutrient-rich FTYPEs (Fig. 1 in Ojanen et al. 2013). Other studies also stress the importance of litter input in the C balance of drained peatland forests, and despite plant-community-level litter decomposition rates being somewhat higher in drained than undrained peatland forests, increased tree litter input can still lead to a significant accumulation of new SOM, especially in conifer-dominated stands (Vávřová et al. 2009, Straková et al. 2012). A model-based assessment of forests growing on mineral soils in Finland further suggests that the proportion of C in litter input that remains in the soil C pool is

ca. 4% (Liski et a. 2006), and although these models are developed for mineral soil forests, the main principles of SOM decomposition also likely apply in drained organic soil forests. Finally, as the maturing drained peatland forests are increasingly harvested, litter production from living trees will inevitably decrease and that from harvest residues increase, making it necessary to explicitly include the input and decomposition of above-ground litter in the calculation method.

Another major change in the estimation of litter input in the new method are the FTYPE-specific turnover rates of arboreal fine roots, which constitute the major source of belowground litter. In the light of recent findings, the assumption of a single turnover rate of 0.85, used for all FTYPEs in the earlier inventory method (Statistics Finland 2021), seems incorrect. The new minirhizotron results included in the new method suggest that tree fine root turnover rate varies markedly by FTYPE and that the turnover rate is on average lower in the nutrient-poor than in the nutrient-rich FTYPEs (Table 3). The lower turnover rate largely explains the remarkably lower arboreal fine root litter production in the new method (Fig. 6).

## 4.4 Evaluating the predicted soil CO2 balance

To evaluate the soil $CO_2$ balance predicted by our method, it can be compared to 1) IPCC Tier 1 emission factors, 2) emission factors applied in other European countries and 3) empirical results on soil $CO_2$ and C balance. As there are no alternative 30-year dynamic emission time series available, the validity of time series dynamics cannot be adequately evaluated here - this would require another comprehensive study. Instead, we will focus on the question if the predicted soil $CO_2$ balance is at feasible level in comparison to emission factors utilized in other GHG inventories and in comparison to results presented in earlier studies.

The default IPCC emission factors for boreal drained organic soils range from 0.25 (95% CI -0.23–0.73) t $CO_2$-C ha$^{-1}$ yr$^{-1}$ in drained nutrient poor forest land to 0.93 (0.54–1.3) t $CO_2$-C ha$^{-1}$ yr$^{-1}$ in nutrient rich forest land (IPCC 2014; Table 2.1 in Chapter 2). Applying the IPCC default emission factors for the areas of nutrient-poor and nutrient-rich FTYPEs found in southern and northern Finland (Table 1) gives emission factors of 0.56 and 0.49 t $CO_2$-C ha$^{-1}$ yr$^{-1}$ for these regions, respectively, and an overall factor of 0.53 t $CO_2$-C ha$^{-1}$ yr$^{-1}$ for the whole country. The emission rate of 0.50 t $CO_2$-C ha$^{-1}$ yr$^{-1}$ predicted by our method for Finland for the year 2021 is slightly lower, but as the emissions with the new method have a rising trend (Fig. 5), they will likely soon exceed the default emissions.

To compare our results to emission factors used for drained organic forest soils by other European countries, we collected information from the national reporting CRF tables published in 2021. Russia, Norway, Ireland and Poland use emission factors of 0.52–0.72 t $CO_2$-C ha$^{-1}$ yr$^{-1}$ that are at the lower end of the 95% CI of IPCC default method emissions for boreal nutrient rich forest land. In temperate areas the IPCC default emission is 2.6 (2.0–3.3) t $CO_2$-C ha$^{-1}$ yr$^{-1}$ and temperate countries Germany, Denmark, Switzerland and UK apply emission factors of 2–2.6 t $CO_2$-C ha$^{-1}$ yr$^{-1}$. As expected, the emissions predicted by our method for Finland in the northern boreal forest zone are lower than the emission factors used in countries that partly or completely locate in the temperate zone.

In a compilation of all available empirical soil $CO_2$ balance data in Finland, Ojanen & Minkkinen (2019) estimated emission factors for nutrient rich (0.66 t $CO_2$-C ha$^{-1}$ yr$^{-1}$) and nutrient poor (-0.19 t $CO_2$-C ha$^{-1}$ yr$^{-1}$) forestry-drained peat soils. Weighting these by the FTYPE areas of our study (Table 1) gives a mean emission factor of 0.37 t $CO_2$-C ha$^{-1}$ yr$^{-1}$. A previous country-level upscaling study by Ojanen et al. (2014) utilizing the heterotrophic soil respiration – litter production method alike the one used in our study (Eqn. 1), gives a maximum value (95 % CI) of 0.55 t $CO_2$-C ha$^{-1}$ yr$^{-1}$ for the mean emission factor. Other soil $CO_2$ balance (Bjarnadottir et al. 2021, Korkiakoski et al. 2023, Minkkinen et al. 2018, Uri et al. 2017, Meyer et al. 2013) and C balance (Minkkinen et al. 1999, Minkkinen & Laine 1999, Simola et. al 2012, Lupikis & Lazdins 2017) studies from boreal and hemiboreal Northern Europe further reinforce the view that forestry-drained peat soils have relatively low $CO_2$ emission; and while drainage has often led to soil C loss, especially nutrient poor soils may exhibit soil C sink even after drainage.

Based on the above paragraphs, we can state that the soil $CO_2$ balance predicted by our new method is in line with previous knowledge and within the range of previous results. Even the relatively large change from close-to-zero emissions on 1990s to close to 10 Mt $CO_2$ year$^{-1}$ emissions in 2021 is within the limits of previous knowledge. According to our uncertainty assessment (Appendix A), which covers sampling error and parameter uncertainty of the original models (Ojanen et al. 2014) and the present GHG inventory time series application, this change in emissions is statistically significant. As it is virtually impossible to produce empirical country-level 30-year soil $CO_2$ emission time series for comparison, a comparison with mechanistic modelling results would be a viable option for future research. Mechanistic models that could couple hydrology and C cycle and were capable for large-scale modelling of forestry-drained peat soils are currently not available, but recent developments in hydrological modelling (e.g. Hökkä et al. 2021, Laurén et al. 2021) appear promising for the development of such models.

**4.5 Critical assumptions in the new method**

It is well established that water table depth (WTD) is the master variable controlling decomposition in drained wetlands (e.g. Silvola et al. 1996, Ojanen et al. 2014, Jauhiainen et al. 2019), and one can argue that WTD should be included in methods that aim at predicting $CO_2$ release from drained peatland soils. In fact, Ojanen et al. (2010), using the data behind our new method, found that regression models between soil respiration and climate variables had higher coefficient of determination, $R^2$ when WTD was included (see Ojanen et al. 2010, Table 4). However, because direct data of WTD, in particular for the time series starting in 1990, is not nationwide available for GHG inventory purposes in Finland, WTD cannot currently be used as a predictor of decomposition. In undrained peatlands, soil $CO_2$ balance fluctuates following annual variation in water saturation and temperature (e.g. Alm et al. 1997, 1999), tree growth is suppressed by anoxia and BA remains low. Draining initiates tree growth and reduces critical high water level periods, and finally, the increasing tree growth and BA lead to lowering water table level (Sarkkola et al. 2010). The impact of WTD on decomposition is therefore implicitly included in

our method through the impact of BA, which provides a proxy of the rate of evapotranspiration that largely controls WTD in forestry-drained peatlands (Hökkä et al. 2021, Leppä et al. 2020).

The *in situ* minirhizotron data behind the arboreal fine root turnover rates that we apply are rare, and the results we use are the first to compare arboreal fine root growth and mortality in different boreal FTYPEs. The measurements carried out with minirhizotrons focused on tree fine roots with diameter ≤0.5 mm, but we extrapolated the results for all arboreal roots ≤2 mm of diameter. Because thicker fine roots have lower turnover rates, our method may overestimate litter input of roots with a diameter of 0.5–2 mm. In terms of soil $CO_2$ balance, this would lead to underestimation of net $CO_2$ emissions. On the other hand, as the fine root turnover rates of dwarf shrubs are higher than those of trees, at least when compared to Scots pine (Minkkinen et al., unpublished data), our method may in this part underestimate the arboreal root litter input and overestimate the net $CO_2$ emissions.

What may impair the use of BA when estimating SOM decomposition is that relatively little data is available of $CO_2$ release from peatland forest soils after clear-cut and regeneration, or after other forest management options such as cuttings applied in continuous-cover silviculture. Mäkiranta et al. (2010) showed that $CO_2$ release from soil decreased after forest clear-cut because of rising water table and, on the other hand, because of the dryness of peat soil surface. Instead, $CO_2$ release from decomposing harvest residues was high, and because of the removed tree stand and suffering ground vegetation, the site acted as a high source of $CO_2$ emissions during the whole study of four years. Similarly, eddy covariance observations in a drained peatland forest showed markedly elevated $CO_2$ emissions at least two years after clear-cutting (Korkeakoski et al. 2019), thus suggesting significant transitory effects of clear-cutting on soil $CO_2$ balance in peatland forests. However, similarly to Mäkiranta et al. (2010) study, a marked share of increased emissions likely resulted from decomposition of fresh harvest residues also in the Korkeakoski et al. (2019) study. While this harvest residue effect is included in our method in both litter input and decomposition (Fig. 7), the temporary effects of tree cut on ground vegetation are not explicitly implemented.

One fundamental assumption in our new method is that the statistical relationships between BA, ground vegetation litter input and SOM decomposition described by Ojanen et al. (2014) for different FTYPEs can adequately predict these associations in forests also under their future successional change. If future forest stands have features that were not present in the studied field sites, this assumption is not valid. This also has relevance in terms of the uncertainty analysis. A strong structural assumption in our new method is that model parameters, including the litter production and turnover rates, do not change in time. If trend-like changes do exist, the reported uncertainties of change estimates are too small (see Lehtonen and Heikkinen 2016 for an illustration). Better understanding of temporal dynamics would be needed to make the assumptions and uncertainty assessments more comprehensive.

Our method also assumes that the spatial covariation in temperature and decomposition, found by Ojanen et al. (2014) among their field sites, is a causal relationship that can be used to describe the effects of increasing temperature on SOM decomposition under climate warming. In our time series, the May–October mean temperature rises by 0.67 °C and the mean annual temperature by 1.2 °C during the GHG reporting period (Fig. 3). This upward trend in temperature increases

decomposition in our method (Table 2) and will reduce the C sink of forest ecosystems in drained peatlands if it is not simultaneously accompanied by increasing nutrient mineralization, plant growth and litter input. It is well established that heterotrophic soil respiration is sensitive to temperature (Silvola et al. 1996, Kätterer et al. 1998, Meyer et al. 2018), and while the response to temperature may somewhat attenuate with time due to a gradual decrease of easily degradable C and thermal adaptation of decomposer communities and their respiration (Davidson and Janssens 2006, Bradford et al. 2008, Bradford 2013), it is likely that the trend of increasing SOM decomposition produced by our method represents a real outcome of climate warming. Globally the same impact of increasing temperature on soil $CO_2$ emissions in drained peatlands is demonstrated by the steeply increasing emission factors from high latitudes to tropics (IPCC 2014; Table 2.1 in Chapter 2).

## 4.6 Further development needs

The impact of WTD on SOM decomposition is implicitly included in our method through BA and T, which both are proxies of the rate of evapotranspiration. However, the relationship between BA and WTD is not linear but attenuating at high BAs (Sarkkola et al. 2010). Furthermore, while ditch spacing and ditch depth have a minor role in comparison to BA, they also affect WTD (Sarkkola et al. 2010, Hökkä et al. 2021), and including models that could link ditch parameters to soil $CO_2$ release would likely improve our method. Models to overcome the shortages of large scale WTD estimation in forest drainages are being tested (e.g. Haahti et al. 2015), but finally resolving these issues may require transforming the method into a process-based model, which would then also open avenues for addressing a wider range of environmental effects on decomposition.

Also, while our new method covers a wide range of litter input, litter input from tree roots thinner than coarse roots, but thicker than fine roots, i.e. in the diameter range of 2–10 mm is lacking. This is because there are no data of the standing biomass or litter production rate for this fraction of tree roots. In a Swedish study, Petersson and Ståhl (2006) showed that the belowground tree biomass was on average 11% higher when roots down to 2 mm diameter were included in comparison to using Marklund's (1987, 1988) biomass equations, which produce the biomass of stump and thicker coarse roots only. Using this estimate, we could assume that we miss ca. 10% of the tree belowground biomass (i.e. of the combined stump and coarse root biomass) and correct the tree belowground biomass estimate accordingly.  However, the turnover rate of this biomass fraction is unknown.

We use 30-year rolling means of climatic variables in our models, computed to the last year of the sequence. This procedure is aimed at smoothing the effects of annual weather variation on the results and thus better reveal the mean level of soil $CO_2$ balance. However, when the smoothed variable has a monotonous trend in time, which is the case, for instance, for May-October or annual mean temperature that increase in time (Fig. 4), the 30-year rolling mean temperatures show past developments behind the present temperature. In our case, this means that the May-October mean air temperature, which largely determines the peat and litter decomposition (Fig. 9 top row), is constantly too low, which in turn leads to too low

$CO_2$ emission estimates. The origin of applying a rolling mean is in the similar solution used in mineral soils. For a more agile smoothed value, another type of filter would be needed.

Finally, although not adopted in our method, peatlands also release organic C as dissolved and particulate organic carbon, DOC and POC into surrounding waters in their natural stage, drainage increases the release (Nieminen et al. 2021). The IPCC 2013 Wetlands Supplement (IPCC 2014) provides default rates of waterborne C release in the form of increased DOC leaching over the natural baseline (IPCC 2014, Ch. 2; Equation 2.5). A simple application of IPCC tier 1 value of 0.12 (0.07–0.19) t C ha$^{-1}$ (IPCC 2014, Ch. 2; Table 2.2), multiplied by the total drained forest land area of 4.3 Mha on organic soil

in Finland, gives a rough estimate of peat C loss as DOC of 0.52 Mt C. Using IPCC default assumption that 90% of DOC is finally released as $CO_2$ to the atmosphere gives an estimate of 1.70 Mt $CO_2$ above the soil $CO_2$ emissions predicted by our new method. This corresponds to a shift of about 21% of the net soil $CO_2$ balance in 2021 (Fig. 5), the amount worthy of considering as part of the GHG inventory of drained peatlands. Although the fate of DOC in water courses is not entirely clear (a significant part of DOC likely ends up to fuel $CO_2$ emissions, but the share of humus that re-sediments to rivers,

lakes and sea is difficult to quantify), from the point of view of peatland forest soils, the runaway organic C reduces the peat C pool and displays one more aspect of the anthropogenic impacts of drainage worth of acknowledging.

## 4.7 Conclusions

Clear merits of our method are that it includes comprehensive empirical data and models for SOM decomposition and litter production and links these to successional changes of tree stand characteristics, monitored by NFI, and to a regionally

precise temperature regime. The method also includes realistic propagation of uncertainties in all model inputs and parameters into the annual estimates of $CO_2$ balance and their differences, which is a requirement for GHG inventories. Such a dynamic model that can consider the effects of both climate change and forest management and development provides a greatly improved tool for forest policy guidance to mitigate climate change.

**Conflict of interest**

The authors declare that they have no conflict of interest.

**Author contribution**

Jukka Alm, Antti Wall, Jukka-Pekka Myllykangas, Paavo Ojanen, Juha Heikkinen, Helena M. Henttonen, Raija Laiho, Kari Minkkinen, Tarja Tuomainen and Juha Mikola have jointly developed the concept of the method described here. J-P Myllykangas is responsible for the development of computations, and all authors have participated in writing and cross-

595 commenting the manuscript.

**Acknowledgements**

The manuscript benefited from expert comments on its earlier version by Professor Chris Evans, UK Centre for Ecology and Hydrology, Dr Andis Lazdiņš, LSFRI Silava, Latvia and Dr Jyrki Jauhiainen, Professor Aleksi Lehtonen, Professor Raisa Mäkipää, Dr Timo Penttilä and Dr Sanna Saarnio, Natural Resources Institute Finland. Dr Riitta Pipatti and Dr Sini Niinistö, Inventory Unit, Statistics Finland, are acknowledged for encouraging the method development process.

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

Figure captions

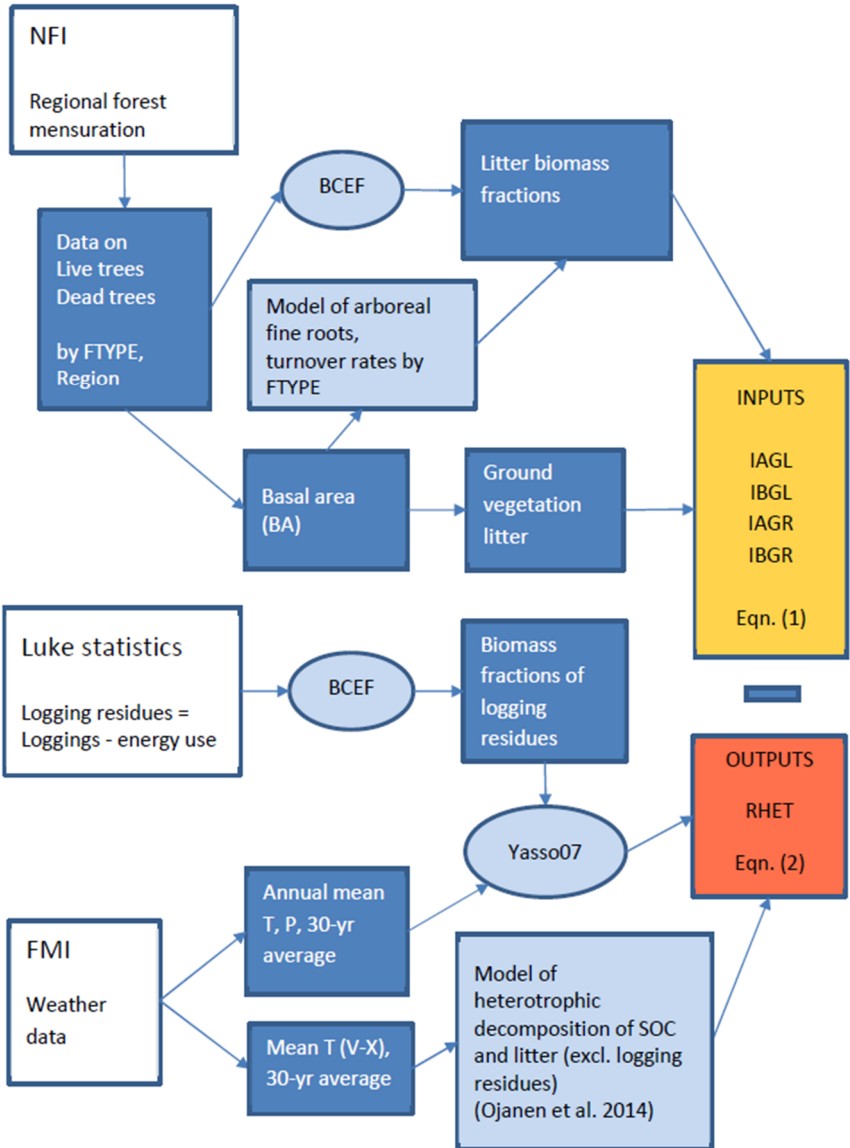

**Figure 1. A conceptual outline to the revised dynamical method to calculate drained peatland forest soil $CO_2$ balance. The arrows denote pathways of information. Sources of monitoring data (white boxes), data used by the dynamic method (dark blue), data conversions and external models (oval), application of original data and decomposition models (light blue) with inputs of C mass to soil (yellow) and outputs of $CO_2$ from soil (red). The minus sign denotes the calculation (Outputs - Inputs) for each annual value of $CO_{2\ Net}$ (Eqn. 1). NFI stands for National Forest Inventory of Finland, Luke Statistics refers to public statistical database of Luke (Natural Resources Institute Finland), FMI is the Finnish Meteorological Institute, FTYPE denotes forest site type categories. T and Pare temperature and precipitation statistics, respectively. BCEF means biomass conversion and expansion factors to proportions of different biomass fractions, Yasso07 is a model used to decompose biomass from harvesting residues and natural mortality. The inputs (I) consist of above ground litter (AGL) and below ground (BGL) components and harvesting residues of both above ground (AGR) and below ground (BGR) materials.**

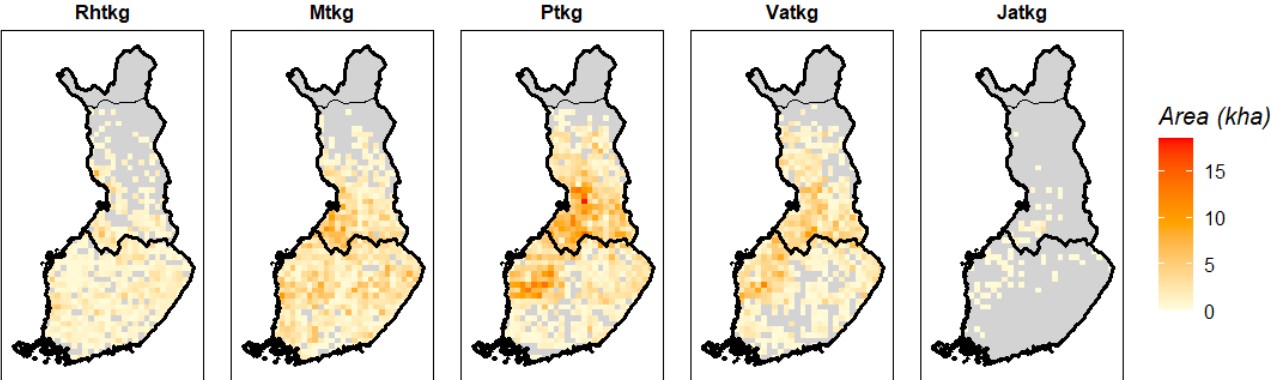

**Figure 2. Distribution of drained peatland forest site types (FTYPEs) in southern and northern Finland, as observed in NFI12. The colour represents the area (kha) of FTYPE in a 20 km × 20 km grid cell and the division of southern and northern Finland (thick black line) follows the division used in the NFI. The northernmost parts of the country (separated with a thin black line) are not included in NFI. Rhtkg = Herb rich FTYPE, Mtkg =** *Vaccinium myrtillus* **FTYPE, Ptkg =** *Vaccinium vitis-idaea* **FTYPE, Vatkg = Dwarf shrub FTYPE, Jatkg =** *Cladonia* **FTYPE, listed in order of decreasing soil fertility.**

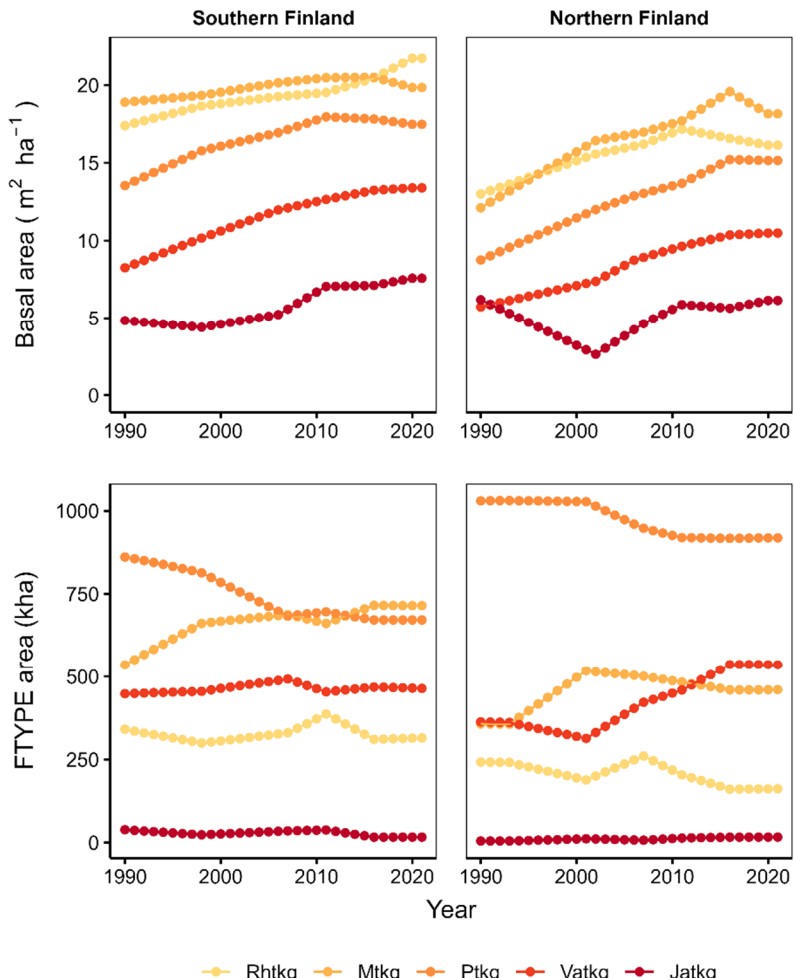

**Figure 3. Time series of basal area per hectare for each FTYPE, and FTYPE areas in southern and northern Finland as estimated from NFI data. htkg = Herb rich FTYPE, Mtkg =** *Vaccinium myrtillus* **FTYPE, Ptkg =** *Vaccinium vitis-idaea* **FTYPE, Vatkg = Dwarf shrub FTYPE, Jatkg =** *Cladonia* **FTYPE, listed in order of decreasing soil fertility.**

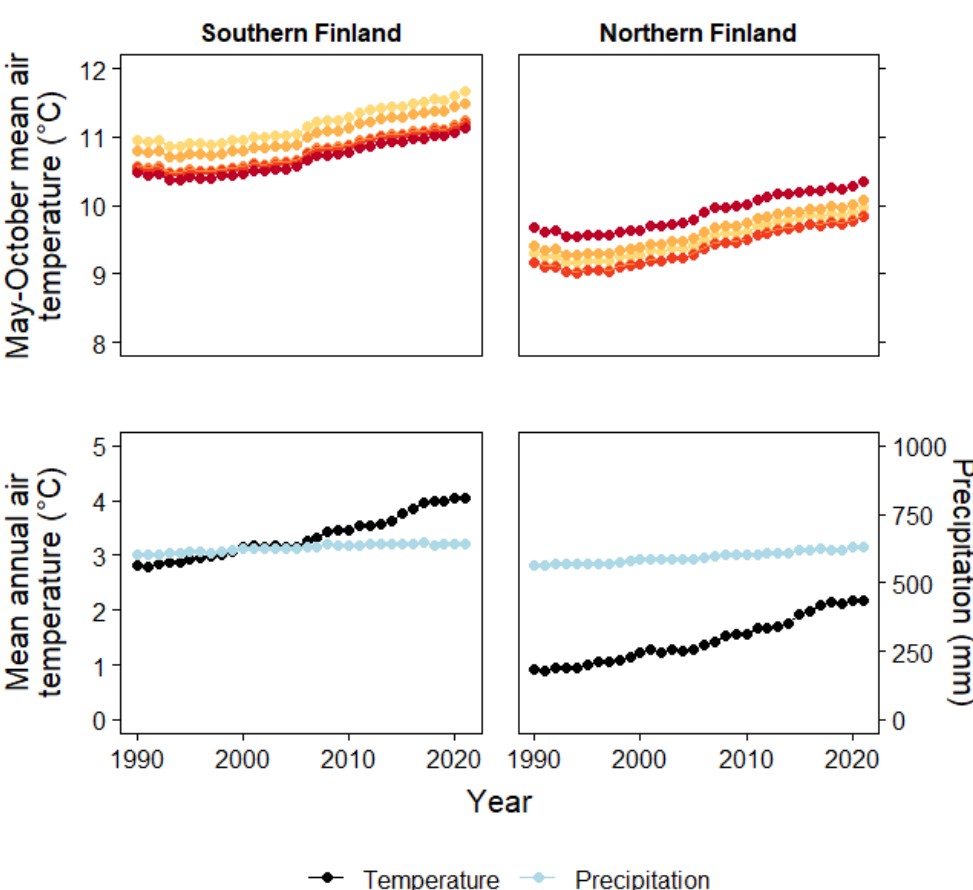

**Figure 4. May–October mean air temperature for each FTYPE in southern and northern Finland and mean annual air temperature and precipitation of all drained peatland forests in southern and northern Finland for 1990–2021. In all cases, the means are 30-yr rolling means. Rhtkg = Herb rich FTYPE, Mtkg = *Vaccinium myrtillus* FTYPE, Ptkg = *Vaccinium vitis-idaea* FTYPE, Vatkg = Dwarf shrub FTYPE, Jatkg = *Cladonia* FTYPE, listed in order of decreasing soil fertility.**

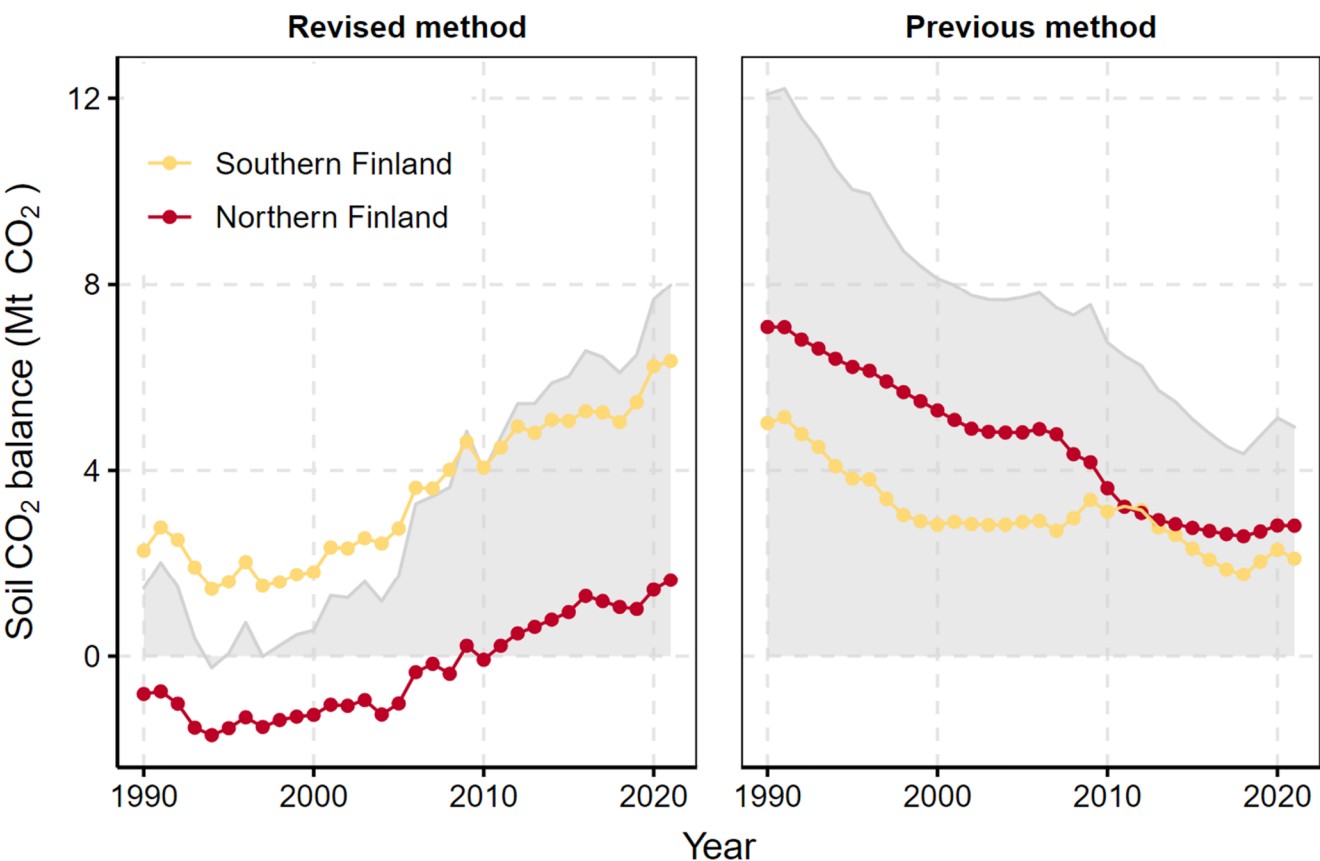

**Figure 5. Drained organic forest soil CO₂ balance for the whole country (grey area) and southern Finland (yellow line) and northern Finland (red line) as produced by the new fully dynamic and the previous semi-dynamic method. See Fig. 1 for the two regions of the country.**


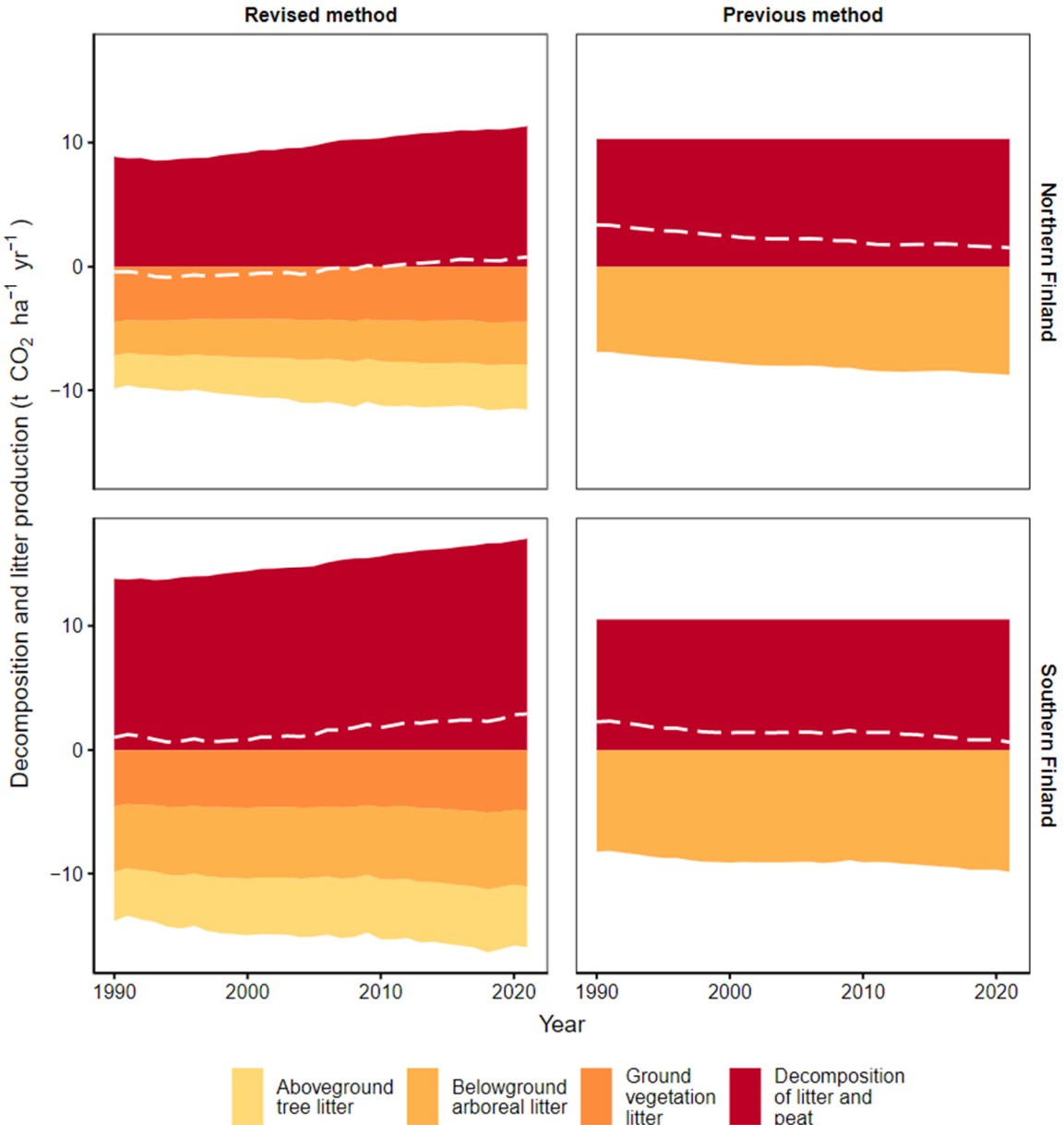

**Figure 6. Input of litter fractions, CO₂ release from decomposing litter and peat, and the CO₂ balance (dashed white line) for southern and northern Finland using the new fully dynamic and the previous semi-dynamic method. In the new method, ground vegetation litter consists of both aboveground and belowground litter except for dwarf shrub fine root litter, which is included in belowground arboreal litter.**

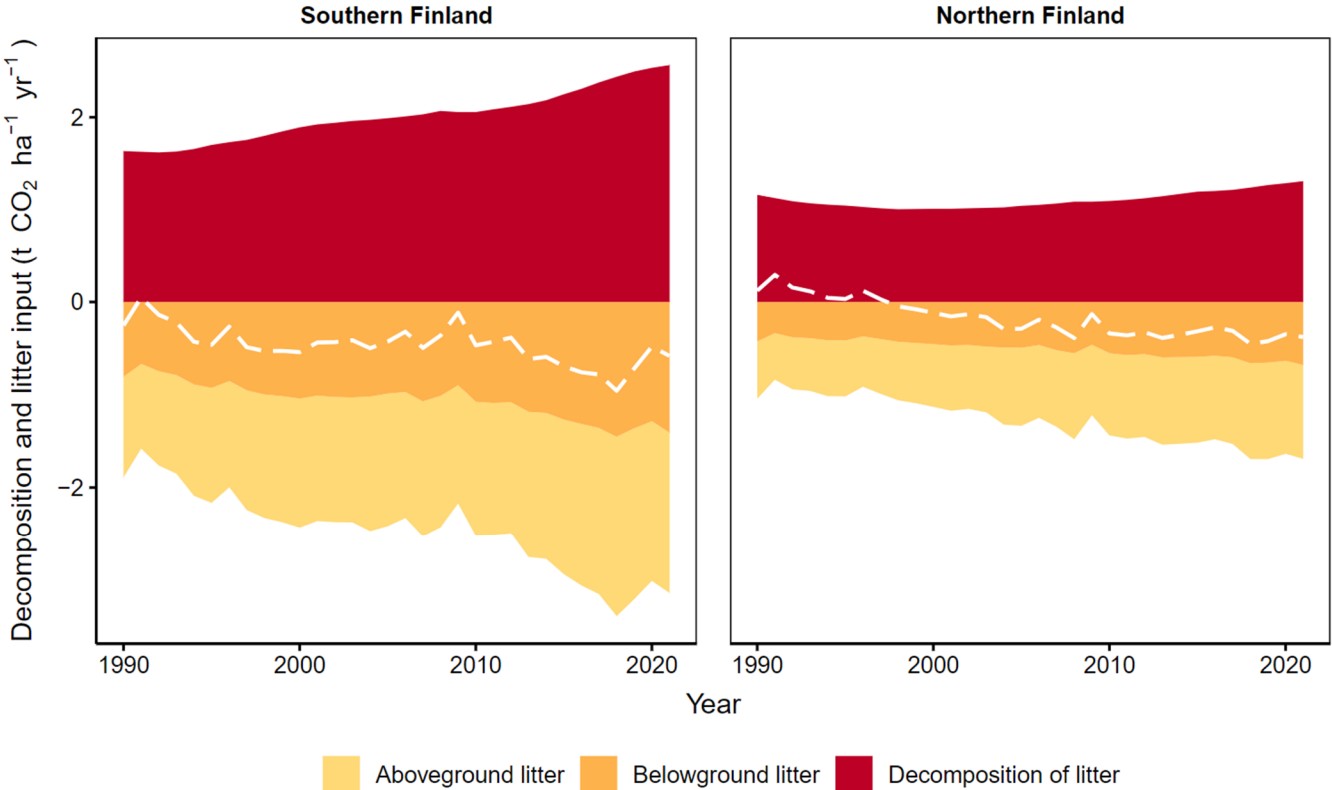

**Figure 7. Input and decomposition of litter from tree harvests and naturally died trees, and their CO$_2$ balance (dashed line) for southern and northern Finland as produced by the new method. Only stem and stump wood are included for naturally died trees.**


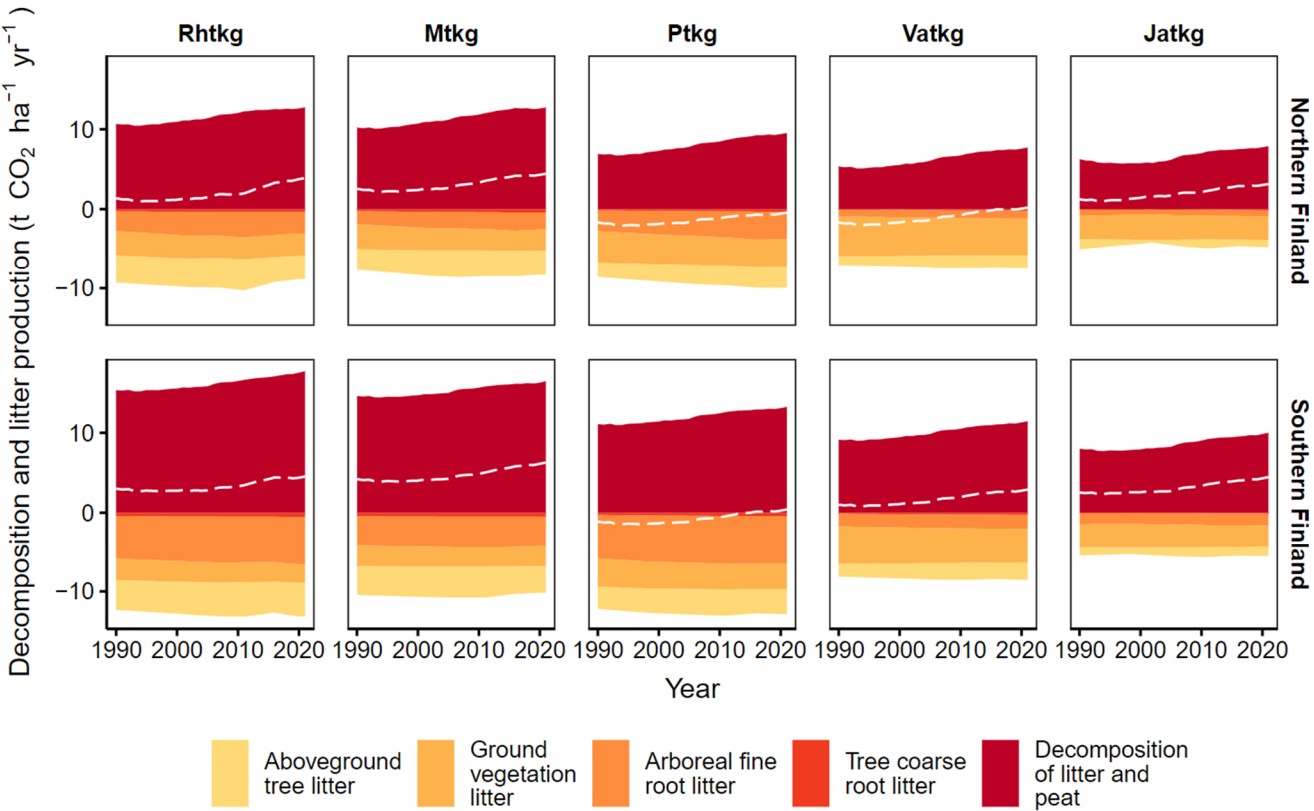

**Figure 8.** Litter input and decomposition (both exclude input and decomposition of litter from harvested and naturally died trees), and the soil $CO_2$ balance (dashed line) in FTYPEs in southern and northern Finland as produced by the new method. Ground vegetation litter includes aboveground and belowground litter except for dwarf shrub fine root litter, which is included in arboreal fine root litter. Rhtkg = Herb rich FTYPE, Mtkg = *Vaccinium myrtillus* FTYPE, Ptkg = *Vaccinium vitis-idaea* FTYPE, Vatkg = Dwarf shrub FTYPE, Jatkg = *Cladonia* FTYPE, listed in order of decreasing soil fertility.


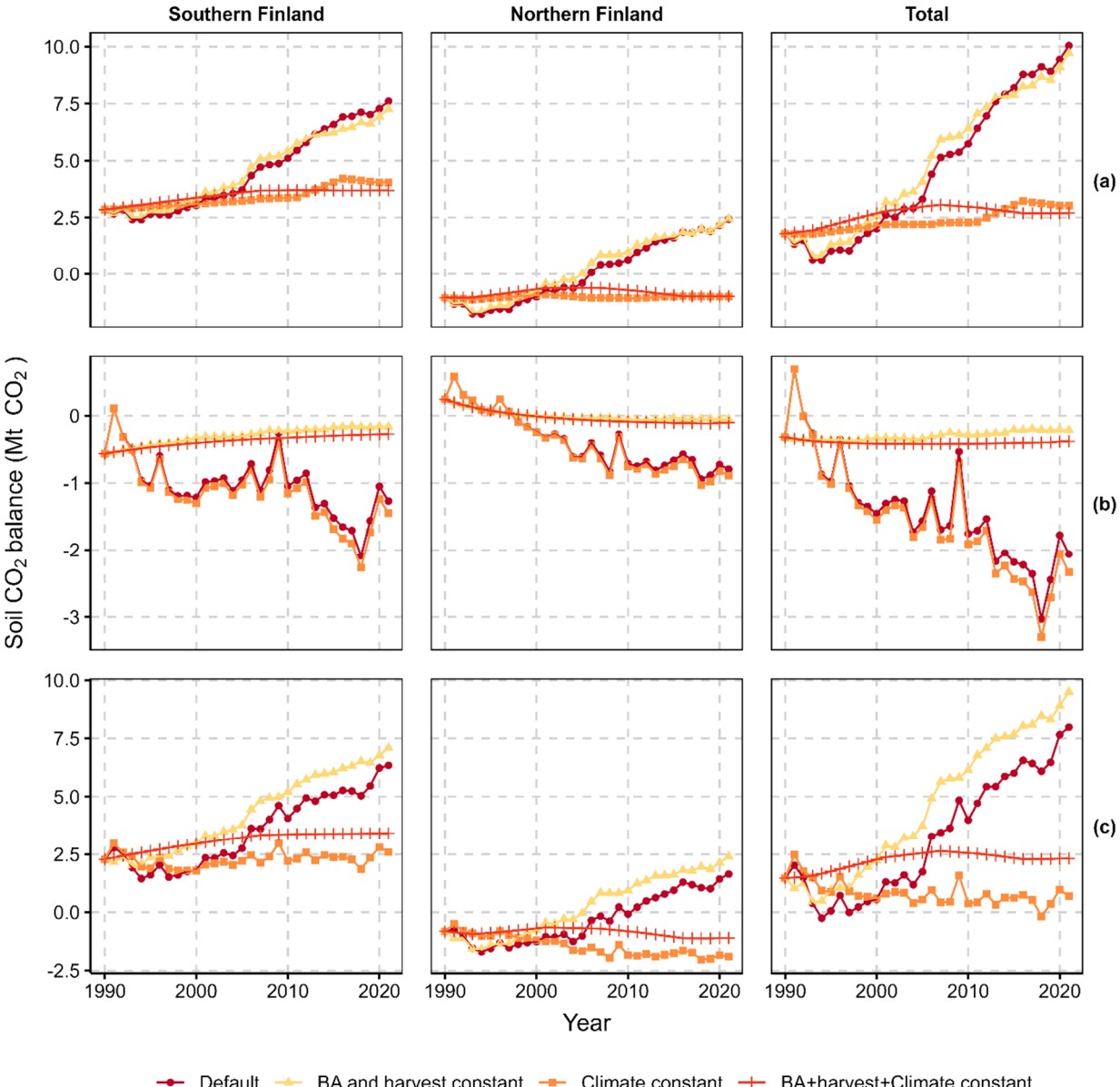

**Figure 9. Soil CO$_2$ balances calculated with four different scenarios for southern and northern Finland and the whole country:**
**"Default" allows all drivers change according to data, "BA and harvest constant" fixes the values of BA and harvest rate to that in 1990, "Climate constant" fixes the growing season mean temperature used by equations of SOM decomposition and the climatic variables used by Yasso07 to values recorded in 1990, and the "BA + harvest + climate constant" fixes all these variables to their 1990 levels. The top row of panels (A) shows the CO$_2$ balance of live tree and ground vegetation litter input and SOM decomposition (thus excluding harvest residues and litter from naturally died trees), the middle row (B) shows the CO$_2$ balance of input and decomposition of litter from harvested and naturally died trees, and the bottom row (C) shows the total soil CO$_2$ balance.**

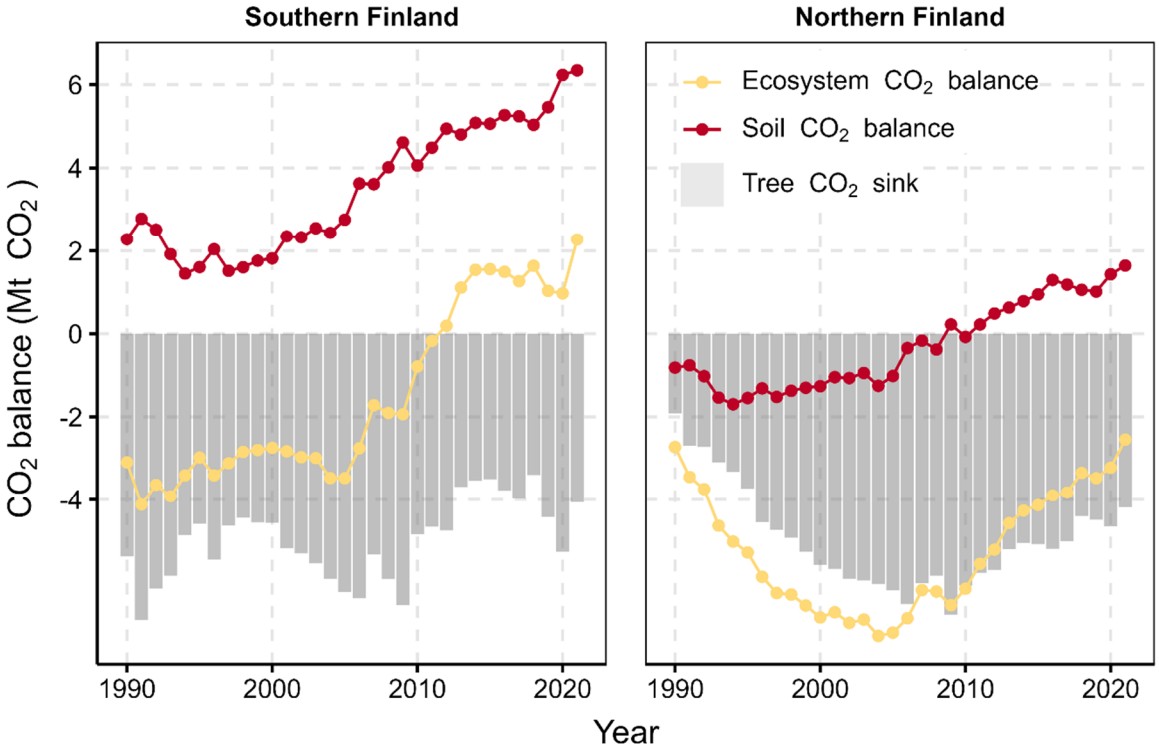

**Figure 10.** Changes in tree $CO_2$ sink (grey bars), soil $CO_2$ balance (red line) and whole ecosystem $CO_2$ balance (yellow line) (Mt $CO_2$ yr$^{-1}$) in drained peatland forests in southern and northern Finland across the GHG inventory reporting period (see Fig. 2 for the two regions). Note that no other GHGs than $CO_2$ is included in the calculated balances.

**Caption for figure in Appendix A**

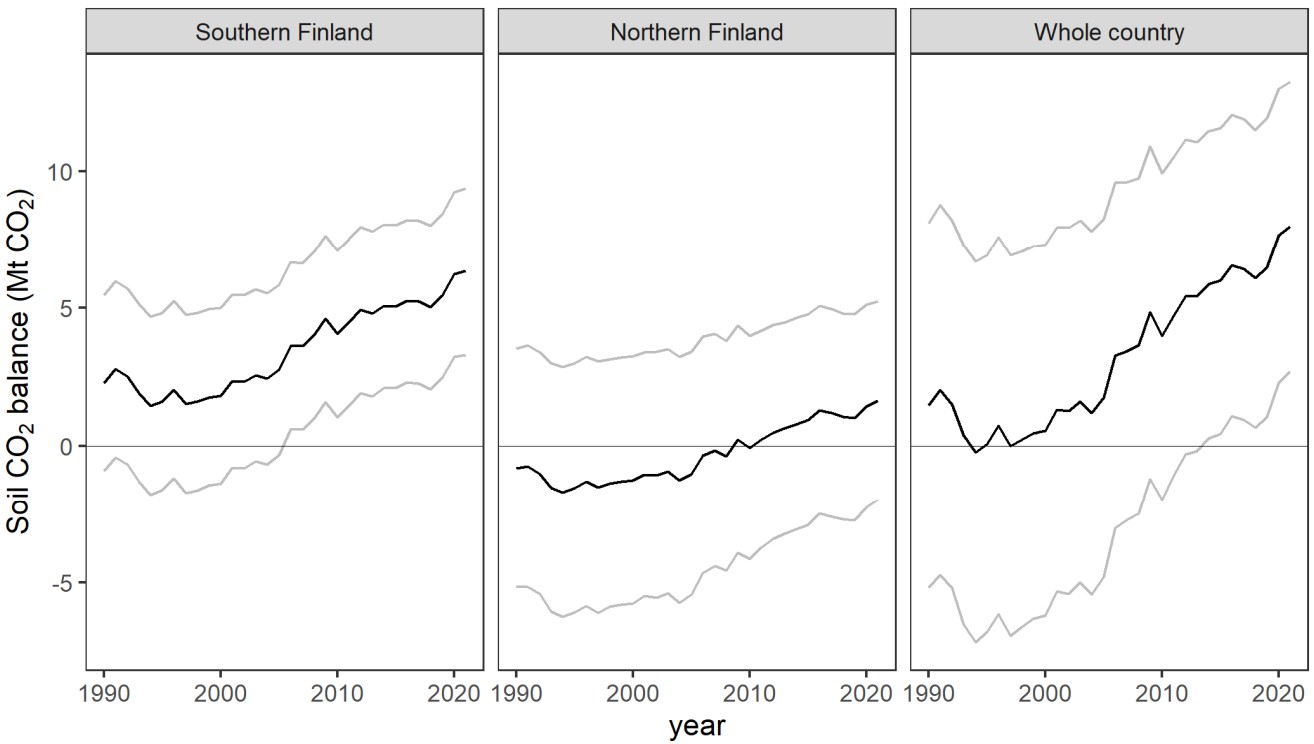

**Figure A1. Drained organic forest soil CO$_2$ balance for southern Finland, northern Finland and the whole country (black lines;**
**equal to those in Figure 4), and pointwise 95% confidence intervals obtained by adding ± U to the balance estimate (grey lines).**

Tables for main text

**Table 1. Characteristics of drained peatland forest site types (FTYPEs) and the GHG inventory estimates of their area and proportion of all drained peatland forest area remaining forest, based on NFI12 data and following the FAO forest classification, in southern and northern Finland (n = number of sample plot centres in each FTYPE). See Fig. 1 for the southern and northern regions of Finland and the distribution of FTYPEs across the regions.**

| FTYPE | Abbreviation | Dominant tree species | Composition of ground vegetation | Southern Finland | Northern Finland |
|---|---|---|---|---|---|
| Herb rich drained peatland forest | Rhtkg | *Betula pubescens, Picea abies* | Herbs with high nutrient requirement such as large pteridophytes, *Oxalis acetocella, Geranium sylvaticum, Cornus suecica, Viola palustris* | 0.315 Mha 14% n = 929 | 0.162 Mha 7% n = 319 |
| *Vaccinium myrtillus* drained peatland forest | Mtkg | *P. abies, Pinus sylvestris* | *Vaccinium myrtillus, Trientalis europaea, Dryopteris carthusiana* | 0.715 Mha 33% n = 2056 | 0.460 Mha 22% n = 949 |
| *Vaccinium vitis-idaea* drained peatland forest | Ptkg | *P. sylvestris* | *V. vitis-idaea, V. myrtillus, Pleurozium schreberi, Dicranum polysetum, Polytrichum commune* | 0.672 Mha 31% n = 1896 | 0.919 Mha 44% n = 1811 |
| Dwarf shrub drained peatland forest | Vatkg | *P. sylvestris* | *Rhododendron tomentosum, V. uliginosum, Empetrum nigrum, P. schreberi, D. polysetum* | 0.465 Mha 21% n = 1314 | 0.536 Mha 26% n = 1052 |
| *Cladonia* drained peatland forest | Jatkg | *P. sylvestris* | Large *Cladonia spp.* patches, *Eriophorum vaginatum, Calluna vulgaris, Sphagnum fuscum,* dwarf shrubs | 0.017 Mha 1% n = 47 | 0.017 Mha 1% n = 40 |
| **Total** | | | | **2.182 Mha n = 6242** | **2.093 Mha n = 4171** |

**Table 2. Empirical regression models of $CO_2$ release from peat and litter decomposition for FTYPEs as used in the new method. The regression models are from Ojanen et al. (2014), except that the constants for the *V. myrtillus* and *V. vitis-idaea* FTYPEs are weighted means of constants of the two subtypes; BA = tree stand basal area ($m^2$ $ha^{-1}$), T = mean May–October air temperature (°C).**

| Drained peatland forest site type | Decomposition (g $CO_2$ $m^{-2}$ $year^{-1}$) |
|---|---|
| Herb rich drained peatland forest | $-1383 + 14.74 \times BA + 242.8 \times T$ |
| *Vaccinium myrtillus* drained peatland forest | $-1440 + 14.74 \times BA + 242.8 \times T$ |
| *Vaccinium vitis-idaea* drained peatland forest | $-1662 + 14.74 \times BA + 242.8 \times T$ |
| Dwarf shrub drained peatland forest | $-1771 + 14.74 \times BA + 242.8 \times T$ |
| *Cladonia* drained peatland forest | $-1814 + 14.74 \times BA + 242.8 \times T$ |

**Table 3. Regression models from Ojanen et al. (2014) for estimating mean arboreal fine root (≤ 2 mm diameter) biomass (g m$^{-2}$) for FTYPEs in southern and northern Finland. BA = mean basal area (m$^2$ ha$^{-1}$); decid = deciduous trees; cover$_{shrub}$ = mean dwarf shrub cover (% of area), taken from Table 3.**

| Region | Arboreal fine root biomass (g m$^{-2}$) |
|---|---|
| Northern Finland | $-53.2 + 8.80 \times BA_{pine} + 6.61 \times BA_{spruce} + 17.3 \times BA_{decid} + 4.81 \times cover_{shrub}$ |
| Southern Finland | $120 + 8.80 \times BA_{pine} + 6.61 \times BA_{spruce} + 17.3 \times BA_{decid} + 4.81 \times cover_{shrub}$ |

**Table 4. Regression models of litter input from ground vegetation (combined dry mass of aboveground and belowground litter excluding dwarf shrub fine root litter) in different FTYPEs. Models are from Ojanen et al. (2014), except for the constants of *V. myrtillus* and *V. vitis-idaea* FTYPEs, which are area weighted means of constants of their subtypes; BA = basal area (m$^2$ ha$^{-1}$).**

| Drained peatland forest site type | Ground vegetation litter input (g m$^{-2}$ year$^{-1}$) |
|---|---|
| Herb rich drained peatland forest | $227 - 4.52 \times BA$ |
| *Vaccinium myrtillus* drained peatland forest | $227 - 4.52 \times BA$ |
| *Vaccinium vitis-idaea* drained peatland forest | $256 - 4.52 \times BA$ |
| Dwarf shrub drained peatland forest | $298 - 4.52 \times BA$ |
| *Cladonia* drained peatland forest | $187 - 4.52 \times BA$ |

**Table 5. Litter production rates of tree biomass components (i.e. the proportion of the component mass that turns into litter in a year) as derived from studies by Lehtonen et al. (2004), Muukkonen and Lehtonen (2004), Starr et al. (2005), Liski et al. (2006) and Ojanen et al. (2014).**

| Tree | Foliage | Dead and alive branches | Stem bark | Stump bark | Coarse roots (ø > 10 mm) |
|---|---|---|---|---|---|
| Pine | 0.33 | 0.02 | 0.0052 | 0.0029 | 0.0184 |
| Spruce | 0.1[S], 0.05[N] | 0.0125 | 0.0027 | 0.0015* | 0.0125 |
| Deciduous | 0.79 | 0.0135 | 0.0029 | 0.0001 | 0.0135 |

[S] Southern Finland, [N] Northern Finland, * Calculated as a mean of pine and deciduous tree estimates as no estimate for spruce was available from literature.


**Table 6. Dwarf shrub cover, obtained from Ojanen et al. (2014), and turnover rate of tree fine roots (≤ 0.5 mm diameter) in FTYPEs. The s.d. are calculated from average median turnover rates.**

| Drained peatland forest FTYPE | Dwarf shrub cover (% of area) | Mean tree fine root turnover rate (1 yr$^{-1}$) | n | s.d. |
|---|---|---|---|---|
| Herb rich drained peatland forest | 7 | 0.8 | 3 | 0.20 |
| *Vaccinium myrtillus* drained peatland forest | 15 | 0.5 | 2 | 0.13 |
| *Vaccinium vitis-idaea* drained peatland forest | 32 | 0.7 | 3 | 0.17 |
| Dwarf shrub drained peatland forest | 45 | 0.2 | 2 | 0.07 |
| *Cladonia* drained peatland forest | 40 | 0.2* | 0 | - |

**\*Value replicated from Dwarf shrub FTYPE as no respective site was measured.**

**Table A1. NFI12 estimates of the areas of drained peatland forest site types (FTYPEs with their Finnish abbreviations) in southern and northern Finland together with their standard errors (s.e.) and relative standard errors (RSE) due to sampling assessed as explained in Korhonen et al. (2021, Supplementary file S1).**

| Region | Drained peatland forest site type | Area, Mha | s.e., Mha | RSE,% |
|---|---|---|---|---|
| Southern Finland | Herb rich FTYPE (Rhtkg) | 0.336 | 0.012 | 3.5 |
| | *Vaccinium myrtillus* FTYPE (Mtkg) | 0.750 | 0.018 | 2.4 |
| | *Vaccinium vitis-idaea* FTYPE (Ptkg) | 0.704 | 0.019 | 2.6 |
| | Dwarf shrub FTYPE (Vatkg) | 0.490 | 0.016 | 3.3 |
| | *Cladonia* FTYPE (Jätkg) | 0.018 | 0.003 | 14.8 |
| Northern Finland | Herb rich FTYPE (Rhtkg) | 0.173 | 0.011 | 6.1 |
| | *Vaccinium myrtillus* FTYPE (Mtkg) | 0.488 | 0.018 | 3.8 |
| | *Vaccinium vitis-idaea* FTYPE (Ptkg) | 0.972 | 0.027 | 2.8 |
| | Dwarf shrub FTYPE (Vatkg) | 0.572 | 0.021 | 3.7 |
| | *Cladonia* FTYPE (Jätkg) | 0.019 | 0.003 | 16.8 |

**Table A2. NFI12 estimates of basal area of trees and their standard errors (s.e.) and relative standard errors (RSE) due to sampling assessed as explained in Korhonen et al. (2021, Supplementary file S1).**

| Region | Drained peatland forest site type | Tree species category | Basal area m$^2$ ha$^{-1}$ | s.e. m$^2$ ha$^{-1}$ | RSE % |
|---|---|---|---|---|---|
| Southern Finland | Herb rich type (Rhtkg) | Pine | 2.80 | 0.21 | 7.3 |
| | | Spruce | 9.80 | 0.36 | 3.6 |
| | | Deciduous | 7.81 | 0.25 | 3.3 |
| | | All species | 20.42 | 0.41 | 2.0 |
| | Vaccinium myrtillus type (Mtkg) | Pine | 6.52 | 0.18 | 2.8 |
| | | Spruce | 8.30 | 0.19 | 2.3 |
| | | Deciduous | 5.63 | 0.13 | 2.4 |
| | | All species | 20.44 | 0.27 | 1.3 |
| | Vaccinium vitis-idaea type (Ptkg) | Pine | 12.64 | 0.17 | 1.4 |
| | | Spruce | 1.85 | 0.08 | 4.3 |
| | | Deciduous | 3.28 | 0.10 | 3.0 |
| | | All species | 17.78 | 0.20 | 1.1 |
| | Dwarf shrub type (Vatkg) | Pine | 12.13 | 0.16 | 1.3 |
| | | Spruce | 0.21 | 0.03 | 11.8 |
| | | Deciduous | 0.89 | 0.05 | 6.1 |
| | | All species | 13.24 | 0.17 | 1.3 |
| | Cladonia type (Jätkg) | Pine | 6.62 | 0.76 | 11.5 |
| | | Spruce | 0.04 | 0.04 | 98.9 |
| | | Deciduous | 0.46 | 0.19 | 41.2 |
| | | All species | 7.12 | 0.79 | 11.0 |
| Northern Finland | Herb rich type (Rhtkg) | Pine | 2.49 | 0.29 | 11.6 |
| | | Spruce | 5.55 | 0.43 | 7.7 |
| | | Deciduous | 8.53 | 0.42 | 4.9 |
| | | All species | 16.57 | 0.62 | 3.7 |

| | | | | |
|---|---|---|---|---|
| Vaccinium myrtillus type (Mtkg) | Pine | 7.05 | 0.23 | 3.2 |
| | Spruce | 5.24 | 0.23 | 4.5 |
| | Deciduous | 6.66 | 0.22 | 3.2 |
| | All species | 18.95 | 0.34 | 1.8 |
| Vaccinium vitis-idaea type (Ptkg) | Pine | 10.06 | 0.15 | 1.5 |
| | Spruce | 1.64 | 0.08 | 4.9 |
| | Deciduous | 3.54 | 0.12 | 3.5 |
| | All species | 15.24 | 0.18 | 1.2 |
| Dwarf shrub type (Vatkg) | Pine | 8.99 | 0.13 | 1.5 |
| | Spruce | 0.40 | 0.04 | 9.8 |
| | Deciduous | 0.98 | 0.06 | 6.5 |
| | All species | 10.37 | 0.15 | 1.4 |
| Cladonia type (Jätkg) | Pine | 5.27 | 0.41 | 7.8 |
| | Spruce | 0.06 | 0.05 | 84.3 |
| | Deciduous | 0.33 | 0.15 | 45.0 |
| | All species | 5.65 | 0.39 | 6.9 |

**Table A3. Covariance matrix $\Sigma_R$ of the parameters of peat and litter decomposition model derived from Ojanen et al. (2014, Table A.5) after combining the Mtkg and Ptkg subtypes.**

| | $\alpha_1$ | $\alpha_2$ | $\beta_{1,\text{Rhtkg}}$ | $\beta_{1,\text{Mtkg}}$ | $\beta_{1,\text{Ptkg}}$ | $\beta_{1,\text{Vatkg}}$ | $\beta_{1,\text{Jätkg}}$ |
|---|---|---|---|---|---|---|---|
| $\alpha_1$ | 31.763 | -156.919 | 1009.484 | 963.02 | 1127.149 | 1394.019 | 1504.165 |
| $\alpha_2$ | -156.919 | 2987.018 | -30191.511 | -29829.44 | -29845.600 | -31328.913 | -30065.011 |
| $\beta_{1,\text{Rhtkg}}$ | 1009.484 | -30191.511 | 330003.711 | 316311.20 | 312471.710 | 322832.399 | 305729.444 |
| $\beta_{1,\text{Mtkg}}$ | 963.020 | -29829.442 | 316311.204 | 318507.28 | 309368.275 | 319328.700 | 302172.953 |
| $\beta_{1,\text{Ptkg}}$ | 1127.149 | -29845.600 | 312471.710 | 309368.28 | 312036.851 | 317750.070 | 301813.094 |
| $\beta_{1,\text{Vatkg}}$ | 1394.019 | -31328.913 | 322832.399 | 319328.70 | 317750.070 | 339519.810 | 316138.347 |
| $\beta_{1,\text{Jätkg}}$ | 1504.165 | -30065.011 | 305729.444 | 302172.95 | 301813.094 | 316138.347 | 338537.417 |

**Table A4. Covariance matrix $\Sigma_G$ of the parameters of the ground vegetation litter model derived from Ojanen et al. (2014, Table A.4) after combining the Mtkg and Ptkg subtypes.**

| | $\alpha_3$ | $\beta_{2,\text{Rhtkg}}$ | $\beta_{2,\text{Mtkg}}$ | $\beta_{2,\text{Ptkg}}$ | $\beta_{2,\text{Vatkg}}$ | $\beta_{2,\text{Jätkg}}$ |
|---|---|---|---|---|---|---|
| $\alpha_3$ | 1.159 | -17.767 | -26.544 | -20.911 | -15.735 | -6.401 |
| $\beta_{2,\text{Rhtkg}}$ | -17.767 | 877.327 | 407.069 | 320.680 | 241.311 | 98.165 |
| $\beta_{2,\text{Mtkg}}$ | -26.544 | 407.069 | 860.277 | 479.085 | 360.510 | 146.655 |
| $\beta_{2,\text{Ptkg}}$ | -20.911 | 320.680 | 479.085 | 550.832 | 284.002 | 115.531 |
| $\beta_{2,\text{Vatkg}}$ | -15.735 | 241.311 | 360.510 | 284.002 | 532.054 | 86.937 |
| $\beta_{2,\text{Jätkg}}$ | -6.401 | 98.165 | 146.655 | 115.531 | 86.937 | 3059.625 |

**Table A5. Covariance matrix $\Sigma_F$ of the parameters of the fine root biomass model of Ojanen et al. (2014, Table A.2).**

| | $\delta_{\text{pine}}$ | $\delta_{\text{spruce}}$ | $\delta_{\text{deciduous}}$ | $\alpha_5$ | $\gamma_{\text{south}}$ | $\gamma_{\text{north}}$ |
|---|---|---|---|---|---|---|
| $\delta_{\text{pine}}$ | 5.321 | 2.056 | 2.162 | 0.202 | -85.607 | -48.226 |
| $\delta_{\text{spruce}}$ | 2.056 | 5.070 | 0.285 | 0.664 | -59.533 | -40.488 |
| $\delta_{\text{deciduous}}$ | 2.162 | 0.285 | 6.011 | 1.023 | -80.610 | -53.993 |

| | | | | | | |
|---|---|---|---|---|---|---|
| $\alpha_5$ | 0.202 | 0.664 | 1.023 | 1.154 | -31.601 | -25.838 |
| $\gamma_{south}$ | -85.607 | -59.533 | -80.610 | -31.601 | 2523.821 | 1505.831 |
| $\gamma_{north}$ | -48.226 | -40.488 | -53.993 | -25.838 | 1505.831 | 1582.462 |

**Table A6.** The applied variances of site-type specific fine-root turnover rates, Var($\phi_p$) (expert judgment), and dwarf shrub coverages, Var($\mu_p$) (Ojanen et al. 2014, Table A.3, after combining the Mtkg and Ptkg subtypes); the corresponding values $\phi_p$ and $\mu_p$ are given in Table 3.

| Drained peatland forest site type | Var($\phi_p$) | Var($\mu_p$) |
|---|---|---|
| Herb rich type (Rhtkg) | $0.1^2$ | 21.00 |
| *Vaccinium myrtillus* type (Mtkg) | $0.1^2$ | 7.65 |
| *Vaccinium vitis-idaea* type (Ptkg) | $0.1^2$ | 6.84 |
| Dwarf shrub type (Vatkg) | $0.05^2$ | 11.00 |
| *Cladonia* type (Jätkg) | $0.05^2$ | 116.00 |

**Table A7.** Relative standard errors (%) of litter production from living trees and from harvests and natural mortality on drained peatlands estimated from NFI11.

| Region | Living trees | Harvests and natural mortality |
|---|---|---|
| Southern Finland | 7.433 | 5.903 |
| Northern Finland | 9.596 | 7.327 |

**Table A8.** Correlations of NFI estimates of litter production from living trees.

| Region | NFI | NFI8 | | NFI11 | |
|---|---|---|---|---|---|
| | | south | north | south | north |
| Southern Finland | 8 | 1.000 | 0.657 | 0.951 | 0.607 |
| Northern Finland | 8 | 0.657 | 1.000 | 0.575 | 0.953 |
| Southern Finland | 11 | 0.951 | 0.575 | 1.000 | 0.539 |
| Northern Finland | 11 | 0.607 | 0.953 | 0.539 | 1.000 |

**Table A9.** Estimates of $CO_2$ release from peat and litter decomposition, net C inputs to soil converted to the units of $CO_2$, and soil balance $CO_2$ balance ("Net") for year 2021 together with the variance ("Var") and uncertainty ("U") of the estimates.

| region | component | | | $CO_2$ | Var | % of Var | | U, % |
|---|---|---|---|---|---|---|---|---|
| Southern Finland | Peat and litter decomposition | | | 31.70 | 0.9922 | 41.17 | | 6.16 |
| | Ground vegetation | | | 6.79 | 0.1135 | 4.71 | | 9.73 |
| | Fine roots | | | 9.49 | 0.9368 | 38.88 | | 19.99 |
| | | deep roots | | | 0.0119 | | 1.27 | |
| | | turnover rates | | | 0.7252 | | 77.41 | |
| | | biomass model | | | 0.1665 | | 17.78 | |
| | | dwarf shrub cover | | | 0.0331 | | 3.54 | |
| | Living trees | | | 7.81 | 0.3370 | 13.99 | | 14.57 |
| | Logg. & nat.mort. | | | 1.27 | 0.0056 | 0.23 | | 11.57 |
| | Site type areas | | | | 0.0131 | 0.54 | | 3.54 |
| | Basal areas | | | | 0.0117 | 0.48 | | 3.34 |
| | Net | | | 6.34 | 2.4099 | 100.00 | | 47.98 |
| | | | | | | | | |
| Northern Finland | Peat and litter decomposition | | | 20.94 | 2.2814 | 67.30 | | 14.14 |
| | Ground vegetation | | | 7.35 | 0.1194 | 3.52 | | 9.22 |
| | Fine roots | | | 5.14 | 0.6347 | 18.72 | | 30.38 |
| | | deep roots | | | 0.0035 | | 0.55 | |
| | | turnover rates | | | 0.2657 | | 41.87 | |
| | | biomass model | | | 0.3340 | | 52.63 | |
| | | dwarf shrub cover | | | 0.0314 | | 4.95 | |
| | Living trees | | | 6.02 | 0.3332 | 9.83 | | 18.81 |
| | Logg. & nat.mort. | | | 0.79 | 0.0034 | 0.10 | | 14.36 |
| | Site type areas | | | | 0.0066 | 0.19 | | 9.69 |
| | Basal areas | | | | 0.0113 | 0.33 | | 12.67 |

| region | component | | | CO$_2$ | Var | % of Var | | U, % |
|--------|-----------|---|---|--------|-----|----------|---|------|
| | Net | | | 1.64 | 3.3900 | 100.00 | | 219.73 |
| Whole country | Peat and litter decomposition | | | 52.64 | 4.1284 | 56.84 | | 7.57 |
| | Ground vegetation | | | 14.14 | 0.4385 | 6.04 | | 9.18 |
| | Fine roots | | | 14.63 | 1.6141 | 22.22 | | 17.02 |
| | | deep roots | | | 0.0283 | | 1.76 | |
| | | turnover rates | | | 0.9910 | | 61.40 | |
| | | biomass model | | | 0.5302 | | 32.85 | |
| | | dwarf shrub cover | | | 0.0646 | | 4.00 | |
| | Living trees | | | 13.83 | 1.0312 | 14.20 | | 14.40 |
| | Logg. & nat.mort. | | | 2.06 | 0.0090 | 0.12 | | 9.01 |
| | Site type areas | | | | 0.0197 | 0.27 | | 3.44 |
| | Basal areas | | | | 0.0229 | 0.32 | | 3.72 |
| | Net | | | 7.98 | 7.2638 | 100.00 | | 66.16 |

**Table A10. Estimates of change from 1990 to 2021 in CO$_2$ release from peat and litter decomposition, net C inputs to soil converted to the units of CO$_2$, and soil balance CO$_2$ balance ("Net") for year 2021 together with the variance ("Var") and uncertainty ("U") of the change estimates.**

| region | component | | | CO$_2$ | Var | % of Var | | U, % |
|--------|-----------|---|---|--------|-----|----------|---|------|
| Southern Finland | Peat and litter decomposition | | | 4.71 | 0.5024 | 74.20 | | 29.47 |
| | Ground vegetation | | | -0.86 | 0.0253 | 3.73 | | 36.17 |
| | Fine roots | | | 0.05 | 0.0588 | 8.68 | | 938.14 |
| | | deep roots | | | 0.0000 | | 0.00 | |
| | | turnover rates | | | 0.0304 | | 51.65 | |
| | | biomass model | | | 0.0268 | | 45.56 | |
| | | dwarf shrub cover | | | 0.0016 | | 2.78 | |
| | Living trees | | | 0.76 | 0.0416 | 6.15 | | 52.66 |
| | Logg. & nat.mort. | | | 0.71 | 0.0106 | 1.57 | | 28.66 |

| Region | Component | Sub-component | | | | | |
|---|---|---|---|---|---|---|---|
| | Site type areas | | | 0.0181 | 2.67 | | 6.49 |
| | Basal areas | | | 0.0203 | 3.00 | | 6.88 |
| | Net | | 4.06 | 0.6771 | 100.00 | | 39.72 |
| Northern Finland | Peat and litter decomposition | | 5.60 | 0.4824 | 72.91 | | 24.29 |
| | Ground vegetation | | -0.44 | 0.0489 | 7.40 | | 99.47 |
| | Fine roots | | 0.97 | 0.0693 | 10.47 | | 52.92 |
| | | deep roots | | 0.0001 | | 0.18 | |
| | | turnover rates | | 0.0154 | | 22.16 | |
| | | biomass model | | 0.0524 | | 75.68 | |
| | | dwarf shrub cover | | 0.0014 | | 1.98 | |
| | Living trees | | 1.57 | 0.0303 | 4.58 | | 21.70 |
| | Logg. & nat.mort. | | 1.04 | 0.0021 | 0.31 | | 8.59 |
| | Site type areas | | | 0.0107 | 1.61 | | 8.25 |
| | Basal areas | | | 0.0180 | 2.72 | | 10.70 |
| | Net | | 2.46 | 0.6616 | 100.00 | | 64.92 |
| Whole country | Peat and litter decomposition | | 10.32 | 1.8323 | 78.47 | | 25.71 |
| | Ground vegetation | | -1.30 | 0.1388 | 5.94 | | 56.30 |
| | Fine roots | | 1.03 | 0.2080 | 8.91 | | 87.17 |
| | | deep roots | | 0.0001 | | 0.07 | |
| | | turnover rates | | 0.0551 | | 26.47 | |
| | | biomass model | | 0.1475 | | 70.93 | |
| | | dwarf shrub cover | | 0.0053 | | 2.53 | |
| | Living trees | | 2.33 | 0.0784 | 3.36 | | 23.53 |
| | Logg. & nat.mort. | | 1.74 | 0.0104 | 0.45 | | 11.48 |
| | Site type areas | | | 0.0288 | 1.23 | | 5.10 |

| | | | | |
|---|---|---|---|---|
| Basal areas | | 0.0383 | 1.64 | 5.89 |
| Net | 6.52 | 2.3349 | 100.00 | 45.96 |

### Appendix A. Assessment of uncertainty

Uncertainty assessments were developed for all annual estimates of soil $CO_2$ balance in Southern Finland, Northern Finland, and whole country, as well as for the estimates of change in balance between years 1990 and 2021. The accounted sources of uncertainty included NFI sampling errors in estimates of the areas of drained peatland forest types and basal area and biomass of trees, and uncertainty about the parameters of the models, litter production and turnover rates, and mean dwarf shrub coverage.

### Notation

Soil $CO_2$ balance, Mt year$^{-1}$ (net emission positive), in year $t$ over forests representing site type $p$ in region $r$ (Southern Finland or Northern Finland) was estimated as $Y_{rpt} = A_{rpt}y_{rpt}$, where $A_{rpt}$ is the interpolated NFI estimate of the area, Mha, of site type $p$ in region $r$ and

$$y_{rpt} = R_{rpt}/100 - 44 \cdot 0.5(G_{rpt}/100 + F_{rpt}/100 + L_{rpt} + H_{rt})/12,$$

is the estimated net $CO_2$ exchange per area unit, Mg ha$^{-1}$ year$^{-1}$. The components of $y_{rpt}$ are

$$R_{rpt} = \alpha_1 B_{rpt} + \alpha_2 T_{rpt} + \beta_{1,p},$$

the prediction of peat and litter decomposition, g $CO_2$ m$^{-2}$ year$^{-1}$, obtained using model (13) of Ojanen et al. (2014, Table A.5),

$$G_{rpt} = \alpha_3 B_{rpt} + \beta_{2,p},$$

the biomass of litter from ground vegetation, g m$^{-2}$ year$^{-1}$, as predicted by model (12) of Ojanen et al. (2014, Table A.4),

$$F_{rpt} = \alpha_4 \phi_p \left( \sum_s \delta_s B_{rpst} + \alpha_5 \mu_p + \gamma_r \right),$$

the biomass of litter from fine roots, g m$^{-2}$ year$^{-1}$, as predicted by model (8) of Ojanen et al. (2014, Table A.2), corrected according to Laiho and Finér (1996) and combined with the site-type specific turnover rates,

$L_{rpt}$, the biomass of litter from living trees (excluding fine roots), Mg ha$^{-1}$ year$^{-1}$, and

$H_{rt}$, the biomass of residual organic matter from tree harvesting and natural mortality after decomposition, Mg ha$^{-1}$ year$^{-1}$. In the equations above, $B_{rpt} = \sum_s B_{rpst}$ is the basal area of trees, m$^2$ ha$^{-1}$, based on NFI, $s$ refers to tree species group, $T_{rpt}$ is the time-smoothed mean May-October air temperature, °C, over sites of type $p$ in region $r$, and the remaining static components of the predictors are based on earlier studies. Model parameters

$$\mathbf{\theta}_R = [\alpha_1 \quad \alpha_2 \quad \beta_{1,\text{Rhtkg}} \quad \beta_{1,\text{Mtkg}} \quad \beta_{1,\text{Ptkg}} \quad \beta_{1,\text{Vatkg}} \quad \beta_{1,\text{Jätkg}}]'$$

are based on Ojanen et al. (2014, Table A.5),

$$\boldsymbol{\theta}_G = [\alpha_3 \quad \beta_{2,\text{Rhtkg}} \quad \beta_{2,\text{Mtkg}} \quad \beta_{2,\text{Ptkg}} \quad \beta_{2,\text{Vatkg}} \quad \beta_{2,\text{Jätkg}}]'$$

on Ojanen et al. (2014, Table A.4), and


$$\boldsymbol{\theta}_F = [\delta_{\text{pine}} \quad \delta_{\text{spruce}} \quad \delta_{\text{deciduous}} \quad \alpha_5 \quad \gamma_{\text{south}} \quad \gamma_{\text{north}}]'$$

on Ojanen et al. (2014, Table A.2). Mean dwarf shrub coverages $\mu_p$ are from Table A.3 of Ojanen et al. (2014), correction $\alpha_4 = 1.043$ is based on Laiho and Finér (1996), and turnover rates $\phi_p$ are reported in this article (Table 3). $q = 5$ is the number of peatland forest site types, and the applied values of the site-type specific parameters are listed in Tables 3-6. For $p \in \{2,3\}$, the values $\beta_{1,p}, \beta_{2,p}, \mu_p$ are area-weighted averages of the values in Ojanen et al. (2014, Tables A.5, A.4, and A.3)

over the two subtypes I and II ($w_{\text{MtkgI}} = 0.615$, $w_{\text{PtkgI}} = 0.6$).

The region-specific annual estimates were $Y_{rt} = \sum_p Y_{rpt}$, country-wide annual estimates, $Y_t = \sum_r Y_{rt}$, and corresponding change estimates, $Y_{r,2021} - Y_{r,1990}$ and $Y_{2021} - Y_{1990}$.

**Uncertainty components**

Uncertainty in estimates $A_{rpt}$ and $B_{rpst}$ due to NFI sampling was based on NFI12 (Tables A1 and A2) relying on the

assumptions that (i) the relative standard errors (RSE) $\text{Var}^{1/2}(A_{rpt})/A_{rpt}$ and $\text{Var}^{1/2}(B_{rpt})/B_{rpt}$ are unchanged across time, i.e., same and equal to the corresponding RSE's in Tables A1 and A2 for all $t$ and (ii) the estimates $A_{rp,1990}$ and $A_{rp,2016}$ are uncorrelated, as well as $B_{rps,1990}$ and $B_{rps,2021}$. Assumption (i) is justified by the relatively constant sampling effort of NFI across time, and assumption (ii) by the fact that the estimates for the two years are completely based on different NFI campaigns.

Uncertainty due to estimation of parameters in models (8), (12), and (13) of Ojanen et al. (2014) was based on covariance matrices $\Sigma_R$, $\Sigma_G$, and $\Sigma_F$ of parameter vectors $\boldsymbol{\theta}_R$, $\boldsymbol{\theta}_G$, and $\boldsymbol{\theta}_F$ (Tables A3 - A5) derived from Tables A.5, A.4, and A.2 of Ojanen et al. (2014).

In particular, $\text{Var}(\beta_{1,\text{Mtkg}}) = w_{\text{MtkgI}}^2\text{Var}(\beta_{\text{MtkgI}}) + w_{\text{MtkgII}}^2\text{Var}(\beta_{\text{MtkgII}}) + 2w_{\text{MtkgI}}w_{\text{MtkgII}}\text{Cov}(\beta_{\text{MtkgI}},\beta_{\text{MtkgII}})$, and so on.

Variances of site-type specific estimates of dwarf shrub coverage, $\text{Var}(\mu_p)$, were similarly derived from those presented in

Table A.3 of Ojanen et al. (2014) and variances of fine root turnover rates, $\text{Var}(\phi_p)$ were based on expert judgement (Table A6), as was that of the deep-root correction $\alpha_4$, $\text{Var}(\alpha_4) = 0.012^2$.

Relative standard errors of litter production estimates from living trees on drained peatland sites were estimated by region from NFI8 and NFI11 as explained in Lehtonen and Heikkinen (2016). For annual results, RSE's from NFI11 were applied to the litter series $L_{rpt}$ across all site types and times (Table A7). When aggregating to the whole country and assessing the

uncertainty in change between years 1990 and 2021, correlations between regions and time points (Table A8), due to applying the same biomass models and litter production rates, were also taken into account.

Relative standard error of litter originating from harvests and natural mortality was estimated similarly from NFI11, and was applied to the residual series $H_{rt}$.

Apart from correlations discussed above, all sources of uncertainty were assumed mutually uncorrelated.

**Impacts of uncertainty components on soil $CO_2$ balance estimates**

Assuming negligible correlation between area estimates $A_{rpt}$, variances of region-specific annual balance estimates $Y_{rt} = \sum_p A_{rpt} y_{rpt}$ due to NFI sampling variance in area estimates $A_{rpt}$ were obtained as

$$\mathrm{Var}_A(Y_{rt}) = \sum_p y_{rpt}^2 \, \mathrm{Var}(A_{rpt}),$$

variances of country-wide balanced estimates as

$$\mathrm{Var}_A(Y_t) = \sum_{r,p} y_{rpt}^2 \, \mathrm{Var}(A_{rpt}),$$

and variances of the change in balance between years 1990 and 2021 as

$$\mathrm{Var}_A(Y_{r,2021} - Y_{r,1990}) = \sum_p \left[ y_{rp,2021}^2 \mathrm{Var}(A_{rp,2021}) + y_{rp,1990}^2 \mathrm{Var}(A_{rp,1990}) \right],$$

and

$$\mathrm{Var}_A(Y_{2021} - Y_{1990}) = \sum_{r,p} \left[ y_{rp,2021}^2 \mathrm{Var}(A_{rp,2021}) + y_{rp,1990}^2 \mathrm{Var}(A_{rp,1990}) \right].$$

To evaluate the impact of NFI sampling variance in basal area estimates $B_{rpst}$, balance estimates were rewritten as

$$Y_{rt} = \sum_p C_{1,rpt} + \sum_{p,s} C_{2,rpst} B_{rpst},$$

where intercepts

$$C_{1,rpt} = A_{rpt} \left\{ \frac{\alpha_2 T_{rpt} + \beta_{1,p}}{100} - \frac{44 \cdot 0.5}{12} \left[ \frac{\beta_{2,p} + \alpha_4 \phi_p (\alpha_5 \mu_p + \gamma_r)}{100} + L_{rpt} + H_{rt} \right] \right\}$$

do not depend on basal area estimates and

$$C_{2,rpst} = A_{rpt} \left\{ \frac{\alpha_1}{100} - \frac{44 \cdot 0.5}{12} \left[ \frac{\alpha_3 + \alpha_4 \phi_p \delta_s}{100} \right] \right\}.$$

Then, for the annual region-specific balance estimates,

$$\text{Var}_B(Y_{rt}) = \sum_{p,s} C_{2,rpst}^2 \text{Var}(B_{rpst}),$$

and the variances for the whole country and for the change are obtained similarly (cf. area estimates above).

Considering uncertainty in the parameters of the peat and litter decomposition model, the balance estimates can also be expressed as

$$Y_{rt} = C_{3,rt} + C_{4,rt}\alpha_1 + C_{5,rt}\alpha_2 - \sum_p C_{6,rpt}\beta_{1,p},$$

where $C_{3,r}$ does not depend on these parameters,

$$C_{4,rt} = \sum_p \frac{A_{rpt}B_{rpt}}{100},$$

$$C_{5,rt} = \sum_p \frac{A_{rpt}T_{rpt}}{100},$$

and

$$C_{6,rpt} = A_{rpt}/100.$$

Thus, the impact of uncertainty in the parameters of the peat and litter decomposition model is

$$\text{Var}_R(Y_{rt}) = \mathbf{c}_{rt}' \Sigma_R \mathbf{c}_{rt},$$

where $\mathbf{c}_{rt} = (C_{4,rt}, C_{5,rt}, C_{6,r,\text{Rhtkg},t}, \ldots, C_{6,r,\text{Jatkg},t})'$. The corresponding variances for the whole country were obtained using coefficients

$$C_{4,t} = \sum_{r,p} \frac{A_{rpt}B_{rpt}}{100}, \quad C_{5,t} = \sum_{r,p} \frac{A_{rpt}T_{rpt}}{100}, \text{ and } C_{6,pt} = \sum_r \frac{A_{rpt}}{100}$$

and for the region-specific change estimates with coefficients

$$C_{4,r} = \sum_p \frac{A_{rp,2016}B_{rp,2016} - A_{rp,1990}B_{rp,1990}}{100} \text{ etc.}$$

Variances $\text{Var}_G(Y)$ and $\text{Var}_F(Y)$ due to parameter uncertainty in ground vegetation and fine-root litter production model were derived from covariance matrices $\Sigma_G$ and $\Sigma_F$ and variances $\text{Var}(\alpha_4)$, $\text{Var}(\phi_p)$, $\text{Var}(\mu_p)$ in a similar manner.

Variances in region-specific annual balance estimates due to uncertainty in litter input from living trees were estimated as

$$\mathrm{Var}_L(Y_{rt}) = \sum_p \left( A_{rpt} \frac{44 \cdot 0.5}{12} \right)^2 \mathrm{Var}(L_{rpt}),$$

where variances $\mathrm{Var}(L_{rpt})$ were based on RSE's in Table A7. When propagating these to the whole country, covariances $\mathrm{Cov}(L_{\mathrm{south},pt}, L_{\mathrm{north},pt})$, based on the NFI11 correlation 0.539 (Table A8), were accounted for, as were covariances between time points when considering the change estimates. For instance,

$$\mathrm{Var}_L(Y_{2021} - Y_{1990}) = \mathbf{c}' \Sigma \mathbf{c},$$

where $\mathbf{c} = (1, 1, -1, -1)'$ and $\Sigma$ is the covariance matrix of $(Y_{\mathrm{south},2021}, Y_{\mathrm{north},2021}, Y_{\mathrm{south},1990}, Y_{\mathrm{south},1990})'$ derived using the correlation matrix of Table A8 and variances based on NFI11 for $Y_{r,2021}$ and NFI8 for $Y_{r,1990}$.

Finally, variances due to uncertainty in litter input from harvests and natural mortality,

$$\mathrm{Var}_H(Y_{rt}) = \left( \frac{44 \cdot 0.5}{12} \sum_p A_{rpt} \right)^2 \mathrm{Var}(H_{rt}),$$

were based on RSE's in Table A7 and on the assumption that the input estimates are uncorrelated across regions and time points. To combine all sources of uncertainty, source-specific variances were simply added up:

$$\mathrm{Var}(Y) = \mathrm{Var}_A(Y) + \mathrm{Var}_B(Y) + \mathrm{Var}_R(Y) + \mathrm{Var}_G(Y) + \mathrm{Var}_F(Y) + \mathrm{Var}_L(Y) + \mathrm{Var}_H(Y).$$

Following IPCC Guidelines (Frey & al. 2006), percentage uncertainties were defined as

$$U_Y = 100 \times 1.96 \sqrt{\mathrm{Var}(Y)}.$$

**Results**

Uncertainty of year 2021 (positive) balance estimates for Southern Finland and whole country are less than 100% (Table A9), which indicates that the soils of drained organic peatlands are statistically significant sources of CO2 according to our model predictions. However, the 2021 balance in Northern Finland does not differ significantly from zero. The increase in emissions between years 1990 and 2021 is significant in both regions and in the whole country (Table A10).

The greatest impact was due to uncertainty in the parameter estimates of the peat and litter decomposition model (41 – 67% of variance in 2021 balance estimates, 73 – 78% in change estimates; Tables A9 and A10). The impact of uncertainty in fine-root model parameters was relatively modest (9 – 10%) in change estimation, but somewhat greater (19 – 39%) in annual estimates. From other sources of uncertainty, only litter input from living trees exceeded 10% impact in any of the estimates. In particular, uncertainty in NFI estimates of site type areas and basal areas of trees had negligible impact on annual balance estimates and contributed less than 5% of the variance of change estimates.

The errors in annual balance estimates are positively correlated, because the same model parameters with same estimation errors were applied throughout the series. For example,

$$\text{Corr}(Y_{2021}, Y_{1990}) = \frac{\text{Cov}(Y_{2021}, Y_{1990})}{\sqrt{\text{Var}(Y_{2021})\text{Var}(Y_{1990})}} = \frac{\text{Var}(Y_{2021}) + \text{Var}(Y_{1990}) - \text{Var}(Y_{2021} - Y_{1990})}{2\sqrt{\text{Var}(Y_{2021})\text{Var}(Y_{1990})}} = 0.9.$$

Due to these correlations, change estimates can differ significantly from zero even if confidence intervals of annual estimates have a large overlap (Figure A1).

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
