# Peer review of "A new method for estimating carbon dioxide emissions from drained peatland forest soils for the greenhouse gas inventory of Finland"

_EGUsphere, 2022_

## Author Response (AR1)

**Editor comments**

Dear authors,
Thank you for your point by point response to the reviewers. Considering the substantial nature of their comments, the manuscript will require major revision and further consideration by the reviewers before it can be published. Please update the manuscript according to the reviewer comments and your responses and then resubmit all necessary documents.

In addition, I have added comments with respect to several of your responses to each reviewer below that will require further revision:

Reviewer 1:
**Answer2:** To justify the lack of evaluation, you have stated that a 30-year data set does not exist. However, why can a shorter-term evaluation (a few years even) not be used as a starting point? Would that not be more appropriate than simply stating long-term data sets do not exist so we cannot do any evaluation work? In the least, there should be a comprehensive summary of the Ojalen et al., papers method evaluation and a quantitative attempt made to demonstrate how the uncertainty/evaluation in those studies relates to the method of this manuscript.

**Editor answer1** (present Answer2 to RC1): We fully revised the subsection 4.4 and added more comparisons to the existing studies and data to convince the reader of the reliability of our new method. Please note that the uncertainty estimation in Appendix A mostly draws from the models and their uncertainties in Ojanen et al. (2014).

**Answer 24:** I don't believe this suggestion adequately responds to the reviewers' comment. I would suggest trying to address this point more specifically, rather than simply rewording as the proposed statement remains somewhat ambiguous.

**Editor answer2**: We have split our answer to 24a and 24b for specific assessment of the two reviewer questions.

Reviewer 2:
**Answer 3:** There is no answer37.

**Editor answer3:** "Answer37" refers to Answer37 to referee 2 (RC2 Answer13 in the Discussion AC's) as they have running numbering here.

**Answer4:** "Currently, the main source of forest data in Finland, the NFI, does not monitor WTD in drained peatland forests, but records several features that indirectly give information of peat moisture conditions." Then why not include a model simulation that includes an estimate of WTD usings these indirect methods? This would at least detail the sensitivity of the model to WTD, even if WTD itself is only an estimate.

**Editor answer4**: We address this question on Answer28

**Answer 13:** And what were the conclusions of Ojanen et al., 2010;2014? Was this a worthwhile pursuit? How much is the uncertainty/accuracy of the model affected by the exclusion of CO2 emissions on shorter timescales?

**Editor answer5:** We address this question in new materials in chapter 4.4. and Editor answer1. The uncertainty analysis in Appendix A mostly deals with the Ojanen et al. 2014 models.

Reviewer 3:
**Answer4:** If the "old" method has not been published, why does it need to be referred to at all? This is confusing the reviewers (and editor), I imagine it will only be worse for readers less connected to the topic.

**Editor answer6** (and RC3 Answer43)**:** The description of the old method, which is now called the earlier semi-dynamic method, is needed to understand the necessity for refining the method and to illustrate the drastic changes the new method brings along in the Finnish GHG inventory results. This is not necessarily so interesting to international readers, but highly valuable for national readers, especially as the old method has not been published and is therefore not easily available for comparison. The rationale for also presenting the results produced by the earlier method is explained on L76-83.

**Answer 10:** I do not deem this an acceptable reason for not assessing uncertainty. Surely some form of uncertainty estimation can be used? Is there not at least replicated data to create a standard deviation? Otherwise what you are saying here is more or less "the data is not reliable enough yet (preliminary and cannot assess uncertainty)". If this is the case, then why is it being used?

**Editor answer7**: We calculated averages from median turnover rates given by the Kaplan and Maier root longevities using the n of the replicates (Table 6).

•

**RC1**: 'Comment on egusphere-2022-1424', Anonymous Referee #1, 17 Jan 2023  reply

**Review report, egusphere-2022-1424**

The paper addresses a very important and timely research question, i.e. the CO2 balance of drained forested peat soils. The CO2 emissions from drained forested peat soil receives currently a very strong attention in many countries. Accurate and precise methods to estimate the soil CO2 fluxes, as well as the full ecosystem level CO2 fluxes are urgently needed.

The overall approach is statistical modelling with the main data input from the national forest inventory and meteorological data and parametrization based on empirical data on litter production as well as decomposition of different main components from dominating plant functional types.

The main results is that forested drained organic soils in Finland represents source of CO2 and that the total soil CO2 emissions from these soils have increased from 1.4 to 7.9 Mt CO2 for the period 1990–2021. Accounting for the entire ecosystem, i.e. also including photosynthesis and calculated for the whole country, forests growing on drained peatlands were a net sink of 0.2 Mt CO2 in 2021, i.e. close to C neutral.

The paper is an important contribution to a science based ground for accessing the land-atmosphere CO2 balance of forested drained organic soils.

**Answer1:** Thank you for the positive evaluation of the importance of our paper and the following comments that greatly help in improving the manuscript.

My major concern is that critical validation of the method is missing. The authors have made an extensive comparison with other emission factors. Still I am missing comparisons with direct measurements for a few example systems representing dominating types and climate settings. It is not an easy task, still urgently needed.

**Answer2:** The reviewer raises an important question on the validity of the proposed method. A method for GHG inventory should produce an accurate 30-year emission time series on country-level. The sole reason why this manuscript was written is that, so far, such a method for forestry-drained peat soils has not existed. To validate the results of our new method, we would need an independent, reliable country-level 30-year time series for comparison. However, the current/previous knowledge consists of Tier 1 and Tier 2 emission factors only and we do not have alternative time series to compare with. We totally agree with the reviewer that such comparison would be valuable; but it is a task that requires another study and a whole article of its own. There are no long measurement time series available, and what we would need are time series for tens of sites to compare the country-level consistency. Here we can only compare to Tier 1 and Tier 2 emission factors to see if the emissions based on our method are comparable to emission factors used in other GHG inventories.

Yet, all this said, it is important to realize that the main components of our method have already been extensively evaluated as the method we propose is based on litter production and decomposition estimation methods that are largely published (as cited in the manuscript). Yasso is also widely used and has been extensively tested and calibrated with large datasets. Finally, the core idea of our method, the estimation of soil $CO_2$ balance as a difference between decomposition and litter production has previously been applied in forestry-drained peatlands in Ojanen et al. 2013 (http://dx.doi.org/10.1016/j.foreco.2012.10.008), Ojanen et al. 2014 (http://dx.doi.org/10.1016/j.foreco.2014.03.049), and Uri et al. 2017a,b (http://dx.doi.org/10.1016/j.foreco.2017.05.023; http://dx.doi.org/10.1016/j.foreco.2017.04.004) and its applicability discussed/tested in Ojanen et al. 2012 (http://dx.doi.org/10.1016/j.foreco.2012.04.027). Thus, we think that even though the method we suggest cannot here be validated as a whole, its components are in general well established and trustworthy.

In the revised manuscript, we fully revised the subsection 4.4 and added more comparisons to the existing studies and data to convince the reader of the reliability of our new method. The text now clearly states what validation has been done, what cannot currently be done and what should/could be done in future studies.

Also the current version of the abstract is hard to digest. It currently require that the reader has read the full article before reading the abstract. Please see detailed comments on the abstract.

**Answer3:** We simplified and reorganized the abstract (detailed replies to comments are given below).

Detailed comments:

L15   Discharge C export needs at least to be considered and potential bias if not included must be discussed. Necessarily not in the discussion section

**Answer4**: We now mention in the abstract that dissolved C export is not included in the calculation method (L23-24). This issue is elaborated in the subsection "4.6 Further development needs" (L569-580).

L15    conceptually I agree that the soil C balance is made up by just by above- and below-ground litter input and heterotrophic (saprotrophic) CO2 respiration. AND possibly also discharge C-export.  It is though important to clarify why autotrophic root respiration is excluded.

**Answer5:** We now better state in the abstract that the soil $CO_2$ balance is made up by heterotrophic $CO_2$-C release from SOM decomposition and litter input (L15). In the Methods section (L106-107) we further clarify that "To focus on $CO_2$ release from SOM, not confounded by autotrophic root respiration, the field plots were trenched (Ojanen et al. 2013)."

16-17   Reformulate. Peatlands drained for forestry release CO2 even if the WT is not change. The change in WT due to drainage and forest ET may change the soil CO2 flux but also non-drained peatlands release soil CO2.

**Answer6:** We revised the abstract to state that drained peatlands are characterized by "higher heterotrophic $CO_2$-C release from faster decomposing soil organic matter (SOM)" (L14-15).

L 20   here it is absolutely necessary that you clarify that the CO2 flux you model emanate from saprotrotrophic CO2 production only. Not stating this explicitly will confuse many readers.

**Answer7:** We now state in the first sentence of the abstract that "the soil $CO_2$ balance is affected by heterotrophic $CO_2$-C release from decomposing SOM". In the methods section we further state: "Autotrophic (root) respiration is not a part of the soil C stock change and is therefore not included in $R_{Het}$." (L106-107).

L 24   do not understand. How is "harvested" trees included?

**Answer8:** We now better explain the calculation of harvesting residues in the abstract (L22-23). The Methods section elaborates on the conversion of logging volumes to harvest residue biomass and exclusion of wood material collected for energy use (L212-220).

L25    what area is the CO2 emission representing? Is it total or per unit area? I would very much prefer first presenting per unit area, e.g. m$^{-2}$ or ha$^{-1}$ and then areal totals. Currently it is very confusing.

**Answer9:** Good point, revised as suggested. Figures 5, 6 and 7 already show the emissions per unit area for the time series.

L25    "1.4 to 7.9 Mt CO2"  You must add time unit, i.e. yr$^{-1}$.

**Answer10:** Units were be completed as suggested.

L27    is this totals for northern and southern Finland or what?`

**Answer11:** That is correct. The emission values and what they represent are now better explained in the abstract.

L28    what about the forest floor PFT´s contribution to CO2 uptake. It can be substantial. If it is not in your data it must me clearly stated that its contribution is so small that it can be neglected, which I really doubt.

**Answer12:** The contribution of forest floor vegetation to autotrophic $CO_2$-uptake is taken into account in the above- and belowground litter input to soil (Tables 3, 4) and therefore, to avoid double counting, cannot be taken into account in plant $CO_2$ sink. The assumption is that ground vegetation growth ($CO_2$ uptake) equals litter production (soil C input), which in the long term is a valid assumption. This litter component is now also listed in the Abstract.

L25-30 this result section is very confusing. I suggest presenting both unit area based estimates (also adding the time unit (yr$^{-1}$)) and areal totals

**Answer13:** Revised as suggested.

L49 In the abstract you state an annual drained peatland soil CO2 flux during 2021 of 7.9 Mt and in the introduction state 3.8 for 2020. Thus you need to be specific in the introduction and clarify that according to method xx the annual peatland soil CO2 during 2020 was 3.8 Mt

**Answer14:** We deleted this sentence for clarity and instead added an opening sentence for the paragraph with the same message: "In Finland peatlands drained for forestry comprise an important category of managed lands." (L43).

L63-63 does these references really refer to saprotrophic CO2 flux, NOT including ANY autotrophic respiration. It is very important that you make this very clear. As "soil CO2 flux" normally includes also the autotrophic root respiration I think it is very important that you make it very clear in all of the text what you actually include.

**Answer15:** Thank you for pointing this out: we removed Couvenberg et al. 2011, who used NEP, and Evans et al. 2021, who used EC tower data, but retained Silvola et al. 1996 and Jauhiainen et al. 2019, and finally, added Ojanen et al. (2010) who used trenching. The fact that our method only considers heterotrophic respiration is now more explicitly specified in the text in several places (lines 65, , 105-107).

L80 "…. by the old method" You must add reference after this statement

**Answer16:** Reference to Statistics Finland (2022)/National Inventory Report (2022) was added, L83.

L 87 C mass input

**Answer17:** Expression was completed as suggested, L91, 94.

L 91 think the sentence "Negative values denote net removal of CO2 from the atmosphere" is confusing. While a forested peatland represent a net removal from the atmosphere depends on the entire system (ecosystem) and not just the soil. I suggest that you instead use something like "Negative values denote net increase of soil C ….. and also suggests that the reference to the atmosphere only is valid when considering the entire ecosystem, not just the soil system.

**Answer18:** Agree, we rephrased the sentence as suggested (L96).

L117 "The areas and proportions of FTYPEs of all drained peatland forests remaining forest in southern and northern Finland," !Something is missing in phrase in italics above

**Answer19:** "Forests remaining forest" is a concept used in the greenhouse gas inventory to describe forests that did not undergo land use change. We now explain this, L125-126.

L258 ??? "uncertainty less than 100 %;" what does this mean, 2% or 98% or what? Reformulate

**Answer20:** We revised the expression: it now says "uncertainty 46%" (L282). Moreover, we added instructions on how the uncertainty is interpreted in the subsection 2.7 Uncertainty analysis (L260-262).

L256 – 264 would very much prefer to have data first presented related to unit area, e.g. ha and then as areal totals. Just having national or regional totals makes it impossible to relate to quantitative data from other sources.

**Answer21:** Agree, we added per unit area emissions in the results where appropriate.

L296 give reference to "Yasso07 modelling"

**Answer22:** We would prefer not adding references to the Results section, the references (Tuomi et al. 2009, 2011) are cited in the Methods section on L155.

L 320-321 for the autotrophic CO2 sink strength you must include also the forest floor vegetation component. If not including you must at least do a sensitivity analysis on how not including that term affects the results.

**Answer23:** See Answer12 above.

L 330-331 the increase in annual temperature is NOT relevant. It is only changes in temperatures above zero (simplifying but much better than referring to annual averages) that

actually affects the production or decomposition. If winter time temperatures are -10 or -4 does not affect either litter production or decomposition. Please refer to only seasonally relevant temperatures. Also differentiate between direct temperature effects and e.g. changes in growing season lengths.

**Answer24a:** We agree and now refer to more relevant mean May-October temperatures in this sentence (L357-359). As the regression models used in our method do not include growing season length as a predictor, it is not meaningful to use the growing season length to interpret the results produced by the method.

L 332 how can the temporal increase in soil $CO_2$ flux be 8.1 when you in the result section state a change from 1.4 to 7.9 Mt over the studied time period?

**Answer24b:** This value is the effect of increasing temperature in the sensitivity scenario when the BA and harvest remain at the level of year 1990. It is higher than the increase in the net soil $CO_2$ balance because increasing BA and harvest rates along the years counteract the temperature effect. This is explained on L332-335 and illustrated in Fig. 9.

**Citation**: https://doi.org/10.5194/egusphere-2022-1424-RC1

**RC2**: 'Comment on egusphere-2022-1424', Anonymous Referee #2, 01 Feb 2023  reply

This paper developed a new method to estimate soil $CO_2$ emissions based on empirical data and models for SOM decomposition and litter production from drained peatland forest in Finland. There are some merits for this study, which also provide new results can be utilized in IPCC.

**Answer25:** Thank you for the views that support the manuscript improvement.

However, I am not convinced by the predicted data at the current stage. The major concerns are lacking validation of the calculated soil $CO_2$ emissions.

**Answer26:** Please, see above our Answer2 to RC1.

The yearly time-scale is also not promising.

**Answer27:** We agree that our method is not capable of following annual and seasonal weather dynamics, but this is not the target of the method. GHG inventories must follow the UNFCCC and IPCC guidance and therefore operate on yearly time scales. In reporting, one year at a time is added to the time series from 1990 onwards (L39-43). To filter out annual weather variations and better reveal the anthropogenic impact of changes in drainage area, BA development and harvests on managed land emissions, the weather time series applied in Finnish GHG inventory are smoothed by 30-year averages. Such averages follow long-term trends, such as gradual warming, but do not bring annual fluctuation in the results. We now state this reasoning explicitly in the methods (L151-154). Please, see also below our Answer37.

Second, the authors claimed that water table depth is the main factor that controls decomposition in drained wetlands. So why not predicting water table depth and then calculate soil $CO_2$ emissions?

**Answer28:** New methods for estimating water table depth (WTD) in drained peatland forests are under development, including process modelling and indirect observations of drainage success, drainage density and ditch condition, but these methods are also challenged by difficulties such as how to validate the estimates or to extend the results to the whole time series needed. Currently, there is no regionally covering WTD monitoring or models that could be used to reconstruct WTD for the inventory time series from 1990 onwards in Finland. However, the NFI records BA, which largely controls WTD in forestry-drained peatlands (Hökkä et al. 2021, Leppä et al. 2020). WTD is thus indirectly involved in our method through BA. When WTD monitoring or application of indirect hydrology observations become realistic and the models can be reliably validated with sufficient coverage in different FTYPEs, more appropriate models for soil $CO_2$ balance can be employed. The model described here is as a crucial improvement to the approach earlier in use in the Finnish GHG inventory. The reasons why we cannot use WTD as a predictor and the role of BA as a WTD proxy are explained in the manuscript on L493-504. How the method could be improved by adding e.g. data of ditch spacing and depth is covered under subsection '4.6 Further development needs' on L547-554.

Some technical comments:

The abstract is hard to understand at the current stage. The authors should improve it.

**Answer29:** Agreed. We have simplified and revised the abstract.

Line (L) 13, explain the meaning of LULUCF.

**Answer30:** This abbreviation for Land Use, Land Use Change and Forestry was dropped from the revised abstract, but is explained in Introduction on L42.

L30, explain GHGI.

**Answer31:** We have replaced this by "GHG inventory" in the abstract as well as elsewhere ("GHG" is explained in the Abstract).

Lines 36-50, the logic of these two paragraphs are a bit of confusion. What do you want to say?

**Answer32:** With these two opening paragraphs (currently L37-53) we explain the reasons for our methodological work, describe and quantify the targeted land area and describe the importance of CO2 emissions and removals in drained peatlands for the Finnish GHG inventory. We edited the paragraphs to clarify our message.

2.1, CO2 should be $CO_2$.

**Answer33:** The typo was corrected.

The space between value and % can be deleted.

**Answer34:** The spaces were removed.

What are the units of equations 1 and 2? What's the difference between carbon balance and $CO_2$ balance?

**Answer35:** We added the units to the adjoining text. Litter input to the soil enters as biomass C, while the emissions from heterotrophic respiration exit as $CO_2$ and the balance can be expressed either in terms of C or $CO_2$.

Figures and Tables should be shown in order.

**Answer36:** Figures and Tables are now in order.

The authors used annual temperature, precipitation, and other climatic data to calculate soil $CO_2$ That would cause large discrepancy between the calculated and actual data. I would suggest the authors calculate the daily data, at least monthly data.

**Answer37:** We agree that for $CO_2$ fluxes short time scale dynamics matter a lot. The earlier work that links $CO_2$ emission rates to daily, and further, to annual emissions provides empirical regression models (Table 6) that use seasonal (May-Oct) temperature as a predictor (Ojanen et al. 2010 and Ojanen et al. 2014). In our method, we use these models. The Yasso07 model (Tuomi et al. 2009, 2011), which has been calibrated with extensive data sets and which we use for estimating the decomposition of harvest residues, in turn uses annual weather data.

"Old calculation method" is not a good name, which can be revised to the name of this method.

**Answer38:** There is no formal name for the earlier method used in GHG inventory in Finland. We now call it in the text as "semi-dynamic" or "previous method", applied in the Finnish GHG inventory until the 2020 inventory (Statistics Finland 2022).

Results, how can you convince the readers if your data do not have any validation?

**Answer39:** Please, see our Answer2 above.

**RC3**: 'Comment on egusphere-2022-1424', Anonymous Referee #3, 12 Feb 2023  reply

**General comments**

The manuscripts present a new way to estimate the $CO_2$ contribution from forested drained organic soils, to be used in national inventories for the reporting to UNFCCC. The method is dynamic and can take into account effects of climate change. This is highly needed as these ecosystems are a high source of emissions for a large number of countries.

The new method uses regressions equations based on a large number of investigations in north and south of Finland and from ecosystems with different fertility. The fertility as determined based on the ground vegetation, that also is monitored in the Finnish monitoring program, is used to estimate the different heterotrophic $CO_2$ flux. Also, the Basal Area is used

in the regression equations to estimate input of C to the soil and as an indicator of the drainage status. Some of the components of the total estimates has been generated by using ecosystem model (Yasso07) as for example the amount of harvest residuals or stumps after harvesting.

The method shows that the Southern areas of forest on drained organic soils are net source of $CO_2$ although with high forest growth. The Northern sites is a sink but may become a source in the future, due to the "coming" temperature increase that is expected. The managing effect as harvesting and the forest residuals form this had a large effect on the net $CO_2$ budget.

The manuscript is a highly valuable contribution for to increase the use of the national inventories and gives a possibility to prognose future changes due to both management effects, but more so by a changing climate.

However, it needs a major revision.

**Answer40:** Thank you for noting the importance of the paper, and for the valuable suggestions to improve the manuscript.

I have not been able to conduct a detail review of the manuscript due to that tables A2 to A10 in the annex is missing. At least in the pdf documents that I could download from the egu website, but I assume that the conclusion is a presented in the discussion. Thus, most of the uncertainties is in the $CO_2$ temperature regressions as shown in Ojanen et al 2014.

**Answer41:** We apologize, the tables were accidentally dropped in conversion to the preprint format. However, as assumed by the reviewer, the main message of the tables was explained in the discussion. The tables are duly included in the revised manuscript.

The material is highly complex, with a lot of abbreviation (not all explained as GHGI line 30), referring to regressions (manly in Ojanen et al 2014) and Yasso07 modelling results, so that the concepts used – that I believe is correct, is unclear and hidden.

The abstract is hard to understand, has to be redone.

**Answer42:** We agree. We have produced a diagram of the different steps of the method to help the reader to follow the calculation. Also, we have simplified and revised the abstract. The abbreviation GHGI was removed and called GHG inventory in the edited manuscript.

It is comparing the new method with the "old" used in Finland and the IPCC default method, this needs to be better described. Is the "old method" published more than in the reporting documents for the national reporting? If the old method is not "published" it has to be presented in the supplement, as the reader needs to be able to compare the underlaying assumptions for the two methods.

**Answer43:** The old method has not been published as a full paper, only as part of the Finnish NIR (National Inventory Report for the UNFCCC), but we agree, the reader needs to know the main differences between the old and new method. For this purpose, we describe the structure and assumptions of the old calculation method on L268-278. Introduction also briefly describes the earlier method (L77-85) with reference to the latest Finnish NIR 2022 (Statistics Finland 2022), and reasons to the need of the present revision.

The underlaying concepts of the new method is not clearly presented. For example, that al autotrophic fluxes are deleted. My suggestion is that one makes a first figure were the different compartments of the $CO_2$, and pools used in the different methods are presented. This should also include the IPCC default method – as the new one also use the C flux from the discharge (stream, lakes etc) from the default one.

**Answer44:** Good point, we will compile a diagram of the different compartments and the links connecting the changes in these for the new method (Figure 1). The GHG inventory only concerns the annual changes in different pools. Thus the soil C inputs comprise of litter from aboveground and belowground parts of trees and ground vegetation. Furthermore, harvesting residues and natural mortality of trees add to the litter input. The outputs comprise of $CO_2$-C losses from decomposition of (old) peat matrix and decomposition of the annual (new) litter input. The empirical regression model for soil heterotrophic respiration ($R_{Het}$) covers the decomposition of both the old and new organic materials, excluding that of harvesting residues and natural mortality, which are therefore covered by Yasso07 modelling.

We do not apply the IPCC default DOC release rates as part of the method, but only raise the DOC issue in Discussion L571-5582. We now clearly mention that DOC-carbon loss is not part of our method, L569.

One of these concepts (assumptions) that are not presented in the manuscript, but is there, is that it will not be any change in ground vegetation biomass over time. The changes **from** the ground biomass is taken into the method, in detail both by above and below litter input to the soil and its effect on heterotrophic soil $CO_2$ emission. The effect of the input from the ground biomass in the low fertile system, is shown by a "moss" driven soil organic matter growth. – I agree on that the ground vegetation biomass can be assumed to be constant during most of the stand rotation. But is this the case after harvesting on the fertile system? There will be a bush/shrub increase that will compensate partly for the organic decomposition, that will be emitted during the first thinning. – I presume that no data is available on this at a national scale, but this aught to be simulated using the Yasson07 model. – You need to argue for that the ground vegetation can be assumed to be in a steady state, this is missing.

**Answer45:** True, dwarf shrub areal cover is assumed to remain static for different FTYPEs (Table 3), but the regression models (Table 4) that predict arboreal fine root biomass (which in turn is used to predict arboreal fine root litter input) have BAs of trees as predictors and can thus follow changes in BA. For ground vegetation litter production (excluding the dwarf shrub belowground litter), the models (Table 5) make use of the relationship between BA and litter production of shrubs, herbs and mosses (Ojanen et al. 2014), thereby leading to changes also in ground vegetation litter production after changes in BA. We now explain this reasoning on L210-212 Overall though, we agree that short-term harvesting disturbance on ground vegetation (either negative or positive) is not explicitly implemented in the present model. This deficiency is discussed with other strengths and vulnerabilities of the method on L513-524.

The overall concept and work holds for the new method, but is not presented as well as you could!

**Answer46:** We hope the diagram that we compiled of the calculation steps (Figure 1) helps in clarifying the method (see Answer42 and Answer44 above).

Nearly all table and figure legends need to be redone. All needed information in understanding a figure should be in the text, not referring to other tables. Thus, the abbreviations for the site fertility have to be presented, with the information on how this is related to site fertility.

**Answer47:** We agree and completed the captions as suggested.

**Detailed comments**

Line 129 the reference is FAO 2018, but the text in the reff is from 2020?

**Answer48:** The reference was corrected to 2020.

Line 179 to 185. The minirhizotron section. Here the effect of "stabilization" after the installation, needs to be discussed as it takes 4 years to have a steady state (Strand et al. 2008 Science). Furthermore, there are no uncertainty values for the determined root turnover rates in Table 3.

**Answer49:** We measured fine root longevity during four years, after one year stabilisation time after the installation of the tubes. It has been stated that minirhizotrons underestimate longevity during short (<3 year) studies (Strand et al. 2008) since the stabilization after installation may take several years. The study sites in Strand et al. 2008 were all on mineral soils and the tubes were installed in position of 45 degrees into the soil. Such an installation requires heavy digging of soil, cutting the roots from a wide area and causing heavy disturbance to the soil ecosystem. In our case, the tubes were installed to peat soil by making a small hole to the peat with a pointed stick and pushing the tube down to the soil. No digging of soil took place. Thus we believe that the disturbance to the soil was actually very small, and one year stabilisation time was enough to start the measurements. This reasoning is now given on L197-200.

The uncertainty estimates are now given in Table 6.

Table 2 What are the units?

**Answer50:** There are no units as the values represent rates in Table 5 (former Table 2), i.e. the proportion of the component mass that turns into litter in a year. This is explained on L182-183, but we also added the explanation to the table title.

Line 258 .. (uncertainty less than 100% ; Table A 10 in Appendix).. Table missing and what do you mean?

**Answer51:** The expression was unclear, we rephrased it (L281-286). Also most of the Appendix tables (A2-A10) were accidentally dropped from the original submission, but are now included in the revised version. We use the IPCC (2006) Guidelines convention, where "uncertainty" is defined as $1.96 \times$ S.E.M. given as a percentage of the estimate. When the uncertainty is less than 100 %, zero is not included within the 95 % confidence limits. This is now explained on L261-263.

Line 272 .. northern and south Finland .. please help the reader by always presenting the data as south in comparison with northern, as mostly done in the manuscript.

**Answer52:** OK, we now use the same order throughout the manuscript.

Line 397 .. (and other climate variables in Yasso07).., what variables? And should not the parameters used in the model be presented in a supplement.

**Answer53:** We assume this comment is targeted to L297. These variables are explained on L222 and shown in Fig. 3 and Supplementary Fig. 2.

Line 425 IPCC 2014 missing in the reference list, should it be 2013?

**Answer54:** The 2013 Wetlands Supplement was published by IPCC in 2014 as listed in References (2013 is part of the name of the document).

---

## Referee Report (RR1)

**egusphere-2022-1424     A new method for estimating carbon dioxide emissions from drained peatland forest soils for the greenhouse gas inventory of Finland**

Jukka Alm, Antti Wall, Jukka-Pekka Myllykangas, Paavo Ojanen, Juha Heikkinen, Helena M. Henttonen, Raija Laiho, Kari Minkkinen, Tarja Tuomainen, and Juha Mikola

The manuscript have improved a lot, but is still not ready for publication.

The main concern is related to figure 1, it seems not to be ready and does not link to the equations used. This as for example the modelled decomposition is only driven by temperature, with no link to the input of organic matter.

What I would like to have, is a figure that shows the "workflow" for the new method, - compared with the old one, as a lot of the discussion is about comparing the two methods. And the old one used by the Finnish reporting has not been published.

Short comments

On line 359 are $CO_2$ not correctly typed.

The reference on line 383 "Suomenselkä and Maanselkä" are missing the year and are not found in the reference list either.

---

## Author Response (AR3)

egusphere-2022-1424

Title: A new method for estimating carbon dioxide emissions from drained peatland forest soils for the greenhouse gas inventory of Finland

Author(s): Jukka Alm et al.

MS type: Research article

Answers to referee statements

Dear Editor,

Please find our answers to refree statements after minor revisions.

Figure 1.

The main concern of the referee was Figure 1, which has now been remade using the suggested approach of showing the data sources and workflow. We also marked those parts of the procedure that have been updated by the new dynamic method. Grey boxes have been added to indicate the characteristics of the previous method applied in the Finnish GHG inventory, in comparison to the present method.

L 359

Typos: $CO_2$-subscripts are now in place.

L 383-384

The expressions for geological formations, major water divides called "Suomenselkä" and "Maanselkä"  were understood as an incomplete literature reference. The expression has been refined so that the context is better understandable.

On behalf of all co-authors,

Jukka Alm